

# Diurnal variation of aerosol optical depth and PM$_{2.5}$ in South Korea: a synthesis from AERONET, satellite (GOCI), KORUS-AQ observation, and WRF-Chem model

Elizabeth Lennartson[1], Jun Wang[1,2], Lorena Castro Garcia[1], Cui Ge[1,2], Greg Carmichael[1,2], Meng Gao[1,2], Jhoon Kim[3], Scott Janz[4]

[1]Department of Chemical and Biochemical Engineering, University of Iowa, USA
[2]Center for Global and Regional Environmental Research, University of Iowa, USA
[3]Department of Atmospheric Science, Yonsei University, S. KOREA
[4]Lab for Atmospheric Chemistry and Dynamics, Code 614, NASA Goddard Space Flight Center, U.S. A.

*Correspondence to*: Elizabeth Lennartson (elizabeth-lennartson@uiowa.edu), Jun Wang (jun-wang-1@uiowa.edu)

**Abstract.** Spatial distribution of diurnal variations of aerosol properties in South Korea, both long term and short term, is studied by using 9 AERONET sites from 1999 to 2017 and an additional 10 sites during the KORUS-AQ field campaign in May and June of 2016. The extent to which WRF-Chem model and the GOCI satellite retrieval can describe these variations is also analyzed. In daily average, Aerosol Optical Depth (AOD) at 550 nm is 0.386 and shows a diurnal variation of 20 to -30% in inland sites, respectively larger than the counterparts of 0.308 and ± 20% in coastal sites. For all the inland and coastal sites, AERONET, GOCI, WRF-Chem, and observed PM$_{2.5}$ data consistently show dual peaks for both AOD and PM$_{2.5}$, one at ~ 10 KST and another ~14 KST. In contrast, Angstrom exponent values in all sites are between 1.2 and 1.4 with the exception of the inland rural sites having smaller values near 1.0 during the early morning hours. All inland sites experience a pronounced increase of Angström Exponent from morning to evening, reflecting overall decrease of particle size in daytime. To statistically obtain the climatology of diurnal variation of AOD, a minimum of requirement of ~2 years of observation is needed in coastal rural sites, twice more than the urban sites, which suggests that diurnal variation of AOD in urban setting is more distinct and persistent. While Korean GOCI satellite retrievals are able to consistently capture the diurnal variation of AOD, WRF-Chem clearly has the deficiency to describe the relatively change of peaks and variations between the morning and afternoon, suggesting further studies for the diurnal profile of emissions. Furthermore, the ratio between PM$_{2.5}$ and AOD in WRF-Chem is persistently larger than the observed counterparts by 30-50% in different sites, but no consistent diurnal variation pattern of this ratio can be found. Overall, the relative small diurnal variation of PM$_{2.5}$ is in high contrast with large AOD diurnal variation, which suggests the large diurnal variation of AOD-PM2.5 relationships, and therefore, the need to use AOD from geostationary satellites for constrain either modeling or analysis of surface PM$_{2.5}$ for air quality application.



## 1 Introduction

Aerosols, both natural and anthropogenic, play an important role in air quality and the climate. Their presence leads to pollution events, and they have a direct and indirect role in modifying the Earth's radiation budget and properties of cloud and precipitation, respectively (Kaufman et al. 2002). Aerosols also lead to acute and chronic health effects due to their small

size and ability to be inhaled through the respiratory track to the lungs' alveoli (Pope et al. 2002). As the world continues to industrialize and increase in population (especially in developing countries), it is imperative to understand and mitigate the effects of pollutants on air quality, climate, and human health, in various spatial and temporal scales.

The United States' Air Quality Index (AQI) is monitored on a daily basis to inform the population on how clean or polluted the air in their local area is. The particulate matter (PM) AQI is calculated from "the ratio between 24-hour averages

of the measured dry particulate mass with the National Ambient Air Quality Standard (NAAQS)" (Wang and Christopher 2003). For clean conditions, the 24-hour average NAAQS for fine particulate ($PM_{2.5}$) must be below 35 μg m$^{-3}$ which is 10 μg m$^{-3}$ higher than the World Health Organization's (WHO) recommendation of 25 μg m$^{-3}$ (EPA 2016; WHO 2006). Due its health implications and crucial role in determining daily air quality levels, it is of utmost importance to effectively monitor and predict 24-hour average $PM_{2.5}$ concentrations.

$PM_{2.5}$ concentrations are typically measured from surface monitors. In the United States, there are roughly 600 continuous (hourly) monitors spread throughout the country and managed by federal, state, local, and tribal agencies (EPA 2008). These monitors provide invaluable information regarding $PM_{2.5}$ levels in 24 hours per day and are not affected by clouds since they are fixed at the surface. However, disadvantages include the fact that they do not represent pollution over large spatial areas. Furthermore, many populated locations in the world do not have a single monitor in their vicinity

(Christopher and Gupta 2010).

To gap fill between monitoring sites and provide estimates at locations around the world in emerging need of surface monitors, recent research has focused on using satellite aerosol optical depth (AOD) to predict ground $PM_{2.5}$ concentrations. An early study by Wang and Christopher (2003) relied on a linear relationship to investigate MODIS AOD and 24-hour average and monthly average $PM_{2.5}$ concentrations. Other efforts have combined the use of satellite AOD with local scaling

factors from global chemistry transport models, columnar $NO_2$, and factors such as the planetary boundary height, the temperature inversion layer, relative humidity, season, and site location (Liu et al. 2005; Liu et al. 2004; Ma et al. 2016; van Donkelaar et al. 2010; Zang et al. 2017; Zheng et al. 2016; Qu et al., 2016). A review by Hoff and Christopher (2009) summarizes that "the satellite precision in measuring AOD is ± 20% and the prediction of $PM_{2.5}$ concentrations from these values is ± 30% in the most careful studies."

Since air quality is often assessed with daily (24 hour) or annual averages of surface $PM_{2.5}$, while polar-orbiting satellite only provides AOD retrieval once per day for a given location (Wang et al., 2016), recent research has integrated AOD from geostationary satellites into the surface $PM_{2.5}$ analysis because a geostationary satellite can provide multiple measurements





of AOD per day for a given a location (Wang et al., 2003a; 2003b), thereby better constraining the diurnal variation of PM$_{2.5}$ for estimating 24-hour average PM$_{2.5}$ (Xu et al., 2015).

Here, we study the diurnal variation of PM$_{2.5}$ and AOD and evaluate such variations for air quality applications by focusing on a six-week long (April – June) air quality campaign in KORUS-AQ and the long-term AERONET sites in South
Korea. The campaign is one of its first kind in east Asia that, through international collaborations, integrated aircraft, surface and satellite data, and air quality models to assess urban, rural, and coastal air quality and its controlling factors. In this study, we first investigate the long-term AOD diurnal variation for various South Korean ground sites and then focus our analysis to the KORUS-AQ AOD diurnal variation as described by chemistry transport model and satellite and surface observations. By centering AOD diurnal variation in our analysis, this study seeks to address the following questions:

1.   What is the climatology of AOD diurnal variation in South Korea, both spatially and spectrally? How long should the ground measurement record be needed to derive the climatology of AOD diurnal variation?

2.   To that degree can AOD diurnal variation be captured by GOCI (a geostationary satellite) and WRF-Chem (a chemistry transport model)?

3.   What is the diurnal variation of surface PM$_{2.5}$? How well is the diurnal variation of PM$_{2.5}$ – AOD relationship
15       captured by WRF-Chem?

The rest of the paper is organized as follows: Section 2 gives a brief overview of previous studies and the motivation for this research. Section 3 details the datasets used in this study, and Section 4 contains the methods and analysis of the study. Section 5 closes the paper with a summary and the main conclusions.

## 2 Background and motivation

### 2.1 AOD Diurnal Variation

The study of AOD diurnal variation dates back to the late 1960s but did not gain momentum until near the turn of the century (Barteneva et al. 1967; Panchenko et al. 1999; Peterson et al. 1981; Pinker et al. 1994). Peterson et al. (1981) found the AOD at Raleigh, North Carolina to have an early afternoon maxima at 13-14 local time during the 1969-1975 study period. Pinker et al. (1994) showed that AOD in sub-Saharan Africa increased throughout the day in December 1987 while
the January 1989 data showed a maxima at 13 local time and minima at 10 and 16 local time. As recent as the early 2000s, the science community agreed that the "diurnal effects are largely unknown and little studied due to the paucity of data…" (Smirnov et al. 2002).

Most diurnal variation of AOD research stemmed from the analysis of aerosol radiative forcing which requires the knowledge of the diurnal distribution of key aerosol properties such as AOD, the single scattering albedo, and the asymmetry
factor (Kassianov et al. 2013; Kuang et al. 2015; Wang et al. 2003b). Two early studies developed an algorithm to retrieve AOD diurnal variation from geostationary satellites over water and showed strong AOD diurnal variation during long-range aerosol transport events; Wang et al. (2003b) used April 2001 hourly data from the GMS5 imager and Wang et al. (2003a)




used half-hourly data in July 2000 from the GOES 8 satellites during the ACE-Asia and PRIDE campaigns, respectively. Consistent with AERONET observations, the GOES 8 retrieval over Puerto Rico showed the dust AOD diurnal variation's noontime minimum and early morning or late afternoon maximum. Subsequent work by Wang et al. (2004) investigated the Taklimakan and Gobi dust regions in China using 1999-2000 AOD data from a nearby airport's sun photometer. They found

a "season invariant" diurnal change of more than ±10% for dust AOD, with larger values in the late afternoon. Their results aligned with similar past studies which found the diurnal variation of dust aerosols to be $\pm$ <5-15% depending on the AERONET site's location and distance from a dust source region (Kaufman et al. 2000; Levin et al. 1980; Wang et al. 2003a). However, on a daily basis, the day-to-day variation of AOD can be distinct, up to 150% and both daily diurnal variation changes and relative departures of AOD from the daily mean are of up to 20% (Kassianov et al. 2013; Kuang et al.

10   2015).

       Overall, research based on limited ground-based observations has shown that on a global and annual scale, the AOD diurnal variation exists, albeit relatively small. On a daily and local scale, AOD diurnal variation is significant which calls upon the need of geostationary satellite measurements for both air quality and climate studies. Newer geostationary satellites may play an important role for the future generation of studying AOD diurnal variation.

**2.2 PM$_{2.5}$ Diurnal Variation**

       In addition to AOD diurnal variation, studies have also investigated the diurnal variation of PM$_{2.5}$. Epidemiological studies focused on the mass, size, spatial and temporal variability, and chemical composition of PM to investigate the complex sources and evolution of aerosols in the atmosphere (Fine et al. 2004; Sun et al. 2013; Wittig et al. 2004). In many of these studies, tracer species of primary aerosols and possible components of secondary organic aerosols were the main focus.

(Edgerton et al. 2006; Querol et al. 2001; Sun et al. 2013; Wittig et al. 2004).

       Regarding diurnal variation of PM$_{2.5}$ mass, studies have found different results for various locations around the world. Querol et al. (2001) used data from June 1999-June 2000 and found Barcelona, Spain's diurnal variation of PM$_{2.5}$ in all four seasons to be characterized by an increase from the late afternoon to midnight. This trend was more pronounced in winter and autumn since these concentrations were higher than their spring and summertime values.

In the United States, early studies have focused on the Los Angeles, Pittsburgh, and general southeast US areas. Fine et al. (2004) chose two sites, an urban one located at the University of Southern California (USC) and a rural one in Riverside, and studied the diurnal variation for one week in the summertime and one week in the wintertime. The USC site had a summer peak in the morning and midday with a winter peak in the morning. The Riverside site experienced a summer peak in the morning and a winter peak in the overnight hours. The winter results were attributed to the boundary-layer temperature

inversion that forms throughout the day over the area. A few years later in Pittsburgh, Wittig et al. (2004) found no clear PM$_{2.5}$ diurnal variability due to the combined effect of particulate matter species being transported to the area versus generated locally. Additionally, they concluded that the daily changes in PM$_{2.5}$ concentrations could be "attributed to the major components of the [particulate] mass, namely the sulfate." Data from the 1998-1999 Southeastern Aerosol and



Characterization Study (SEARCH) was used by Edgerton et al. (2006) at four pairs of urban-rural sites. They established the following three main PM$_{2.5}$ temporal variation patterns: large values of > 40-50 µgm$^{-3}$ that occurred on time scales of a few hours, buildup occurring over several days and then returning to normal levels, and peaks during the summer of similar magnitude as the monthly or quarterly averages. Their four sites had similar diurnal variations characterized by maxima at 6-8 a.m. local time and again from 6-9 p.m., similar to those results found by Wang and Christopher (2003) at seven sites in Alabama.

PM$_{2.5}$ concentrations can significantly vary on relatively short time scales, and in order to understand the potency and effects of the individual chemical species and PM$_{2.5}$ as a whole, it is emergently needed to continue to improve the means by which these measurements are taken, increase the amount of long-term measurements available, and investigate other methods that can be used to assist with characterizing PM$_{2.5}$ concentrations and diurnal variation.

**2.3 Diurnal Variation of AOD-PM$_{2.5}$ Relationship**

Recently, studies have focused on using satellite measurements of AOD in order to predict ground-level PM$_{2.5}$ concentrations in addition to investigating the diurnal variations of both components. An early study examined how well MODIS AOD correlated with 24-hour average and monthly average PM$_{2.5}$ using a linear relationship and found linear correlation coefficients of 0.7 and 0.9, respectively. Additionally, when the linear relationship used the 24-hour average PM$_{2.5}$ concentrations, MODIS AOD quantitatively estimated PM$_{2.5}$ AQI categories with an accuracy of 90% (in terms of capturing AQI variability) in cloud-free conditions (Wang and Christopher 2003).

Other efforts have used satellite AOD in combination with additional factors to improve the prediction of ground-level PM$_{2.5}$, such as the planetary boundary layer height, relative humidity, season, and the geographical characteristics of the monitoring sites (Liu et al. 2005; Wang et al. 2010). Similarly, Gupta et al. (2006) found a strong dependence on aerosol concentration, relative humidity, fractional cloud cover, and the mixing layer height when analyzing the relationship between MODIS AOD and ground-level PM$_{2.5}$. They concluded the importance of local wind patterns for identifying the pollutant sources and overall, high correlations can occur for the following four conditions: cloud-free, low boundary layer heights, AOD larger than 0.1 and low relative humidity.

Xu et al. (2015) used GOCI AOD in cloud-free days and a global chemistry transport model (GEOS-Chem) to find significant agreement between the derived PM$_{2.5}$ and the ground measured PM$_{2.5}$ for both the *annual* and *monthly* averages over eastern China. Incorporating AOD data from GOCI, a geostationary satellite, provided improvement for GEOS-Chem, a global chemistry transport model, to estimate ground-level PM$_{2.5}$ for a highly populated and polluted region of the world on a fine spatial resolution. When comparing their results to MODIS AOD derived PM$_{2.5}$, they found better agreement using their model with an R$^2$ value of 0.66. However, in their study, only *daytime-averaged* AOD from both GEOS-Chem and GOCI AOD are used as their study concerned about *monthly and annual* scale.

Hence, one common theme throughout most of the past research, with the exception of Xu et al. (2015), is the use of AOD data from low-earth orbiting (LEO) satellites to establish the AOD- PM$_{2.5}$ relationship. However, all the studies have



relied on the model-simulated diurnal variation of AOD-PM$_{2.5}$ relationship to converting satellite-based AOD (often once per day) to surface PM$_{2.5}$. An integrated analysis of diurnal variation of AOD, PM$_{2.5}$, and their relationship, with unprecedented observations from field campaigns is overdue.

## 3 Study Area and Data

5 Over the last 40 years, South Korea has experienced an extensive list of air quality advances for constituents such as lead, sulfur dioxide and PM$_{10}$ (Ministry_of_Environment 2016). In 2015, they introduced a standard on PM$_{2.5}$, and due to its recent implementation, the historical trends of PM$_{2.5}$ are not well established. Han et al. (2011) suggests that prior to 2005, PM$_{2.5}$ annual averages either increased or remained constant in rural Chuncheon, South Korea. However, there has been a gradual decrease in the annual averages in Seoul since 2005, although the decreases have not been continuous (Ahmed et al. 2015; 10 Ghim et al. 2015; Lee 2014). Annual averages have ranged from 33.5 $\mu$g m$^{-3}$ in 2004 to 21.9 $\mu$g m$^{-3}$ in 2012, but the most recent patterns have been difficult to interpret (Ahmed et al. 2015). Also, since the PM$_{2.5}$ concentrations discussed above were from research studies based only in the Seoul region, the aforementioned findings may not be fully applicable to South Korea as a whole. For this study, we will use all the AERONET data collected over South Korea since early 1990s, as well as rich data sets collected during KORUS-AQ including additional 10 AERONET sites, GOCI data, and WRF-Chem 15 modeling data.

### 3.1 AERONET

The AERONET sites provide "long-term, continuous, and readily accessible" aerosol data, with AOD and the Angström Exponent being two of the available parameters from the direct sun measurements. Sequences are made in eight spectral bands between 340 nm and 1020 nm while the diffuse sun measurements are made at 440 nm, 670 nm, 870 nm, and 1020 nm 20 (Holben et al. 1998). The Version 2, Level 2 quality level data are used for this study which implies that the data are cloud-screened and quality-assured following the procedures detailed in Smirnov et al. (2000). To compare the AERONET AOD values to those commonly used by other data platforms such as satellites and models, the AOD at 550 nm is calculated by using corresponding Angström exponent derived from AOD at 440 nm and 870 nm.

At the time of last access to the AERONET database in July 2017, the stations listed in Table 1 had Version 2, Level 2 25 data available. These stations were further grouped into the following four land classifications: coastal urban, coastal rural, inland urban, and inland rural. Each site's classification membership is represented in Figs. 1a and 1b by its marker. The same marker color schemes are also used in other figures to correspondingly denote AOD diurnal variation at each individual site.



### 3.2 WRF-Chem

The data from joint Weather Research and Forecasting with Chemistry (WRF-Chem) V3.6.1 model between the University of Iowa and NCAR is used. WRF-Chem is a regional meteorology-chemistry model capable of simulating both the chemical and meteorological phenomena within the atmosphere at flexible resolutions to assist with air quality forecasts. It is a fully coupled "online" model in which both the air quality and meteorological components use the same transport scheme, timestep, grid, and physics schemes for subgrid-scale transport. Additionally, aerosol-radiation-cloud interactions are considered (Grell et al. 2005).

The University of Iowa's WRF-Chem forecast for KORUS-AQ provides simulations of meteorology and atmospheric composition at 20 km and 4 km resolution, respectively. A reduced hydrocarbon chemistry mechanism (REDHC) is added to expedite computations, and a simplified secondary organic aerosol formation scheme is added to the sophisticated MOSAIC aerosol module. In the vertical, there are 53 layers distributed between the surface and 50 hPa. The bottom layer closest to the surface has a thickness of ~ 50 m.

To support the KORUS-AQ field campaigns, a Comprehensive Regional Emissions inventory for Atmospheric Transport Experiments (CREATE) was developed based on GAINS-Asia emissions with updated national data. It has 54 fuel classes, 201 sub-sectors and 13 pollutants, including sulfur dioxide ($SO_2$), nitrogen oxides ($NO_x$), carbon monoxide (CO), non-methane volatile organic compounds (NMVOCs), ammonia ($NH_3$), organic carbon (OC), black carbon (BC), $PM_{10}$, $PM_{2.5}$, carbon dioxide ($CO_2$), methane ($CH_4$), nitrous oxide ($N_2O$) and mercury (Hg). The emissions of NMVOCs from vegetation are calculated online using the MEGAN (Model of Emissions of Gasses and Aerosols from Nature) model, which are influenced by land use, temperature, and etc. (Guenther et al. 2006). Biomass burning emissions are taken from the GFED dataset, which is coupled to a plume-rise model (Grell et al. 2011). Sea salt and dust are calculated using Gong et al. (1997) and GOCART (Zhao et al. 2010) schemes, respectively. Chemical initial and boundary conditions are obtained from the Monitoring atmospheric composition and climate (MACC) global tropospheric chemical composition reanalysis.

WRF-Chem data are retrieved according to the latitude and longitude of each of the 15 AERONET sites that were active during the KORUS-AQ timeframe (Table 1). Similar to AERONET, the AOD at 550 nm is calculated. The AOD derived in WRF-Chem is for multiple different observational wavelengths and is based on Angström Exponent relations (Gao et al. 2016; Schuster et al. 2006). The WRF-Chem data are available from 01 May 2016 00:00 UTC – 01 May 2016 11:00 UTC and 02 May 2016 00:00 UTC – 15 June 2016 23:00 UTC.

### 3.2 GOCI

The Geostationary Ocean Color Imager (GOCI) onboard the Communication, Ocean, and Metrological Satellites (COMS) is the world's first geostationary ocean color observation satellite. Launched in 2010, it provides spatial coverage of 2,500 x 2,500 km in northeast Asia at a 500-m spatial resolution. The domain is comprised of sixteen image segments (with resolution of 4 km x 4 km) and the Korean peninsula sits in the center of it. GOCI has six visible and two near-infrared



(NIR) bands at 412, 443, 490, 555, 560, and 680 nm and 745 and 865 nm, respectively (Choi et al. 2012). Observations are taken eight times per day from 00:30 to 07:30 UTC (09:30 to 16:30 KST) (Choi et al. 2017).

To retrieve hourly aerosol data the GOCI Yonsei Aerosol Retrieval (YAER) algorithm was prototyped in 2010 by Lee et al. (2010) and further developed into its Version 1 (V1) form in 2016 by Choi et al. (2016). The aerosol properties from the

retrieval include AOD at 550 nm, fine-mode fraction at 550 nm, single-scattering albedo at 440 nm, Angström Exponent between 440 and 860 nm, and aerosol type.

Recently, the V1 algorithm has been enhanced to the Version 2 (V2) algorithm and now includes near-real-time (NRT) processing and improved accuracy in cloud masking, determination of surface reflectance, and selection of surface-dependent retrieval schemes. The V2 algorithm has shown similar AOD at 550 nm as Moderate Resolution Imagine

Spectroradiometer (MODIS) and Visible Infrared Imaging Radiometer Suite (VIIRS). When validating with AERONET AOD from 2011 to 2016, the V2 reduced median bias and provided an accuracy range of $0.15\tau + 0.05$ compared to V1, where $\tau$ is the AERONET AOD value (Choi et al. 2017). For this research, the GOCI YAER V2 AOD at 550 nm data are downloaded for the KORUS-AQ campaign from 01 May 2016 to 12 June 2016.

### 3.2 PM$_{2.5}$

PM$_{2.5}$ data are downloaded from the KORUS-AQ Data Repository for the entirety of the field campaign. The 10 sites with a corresponding AERONET site nearby are used for this research (Table 2). The three sites of Busan, Gwangju, and Seoul have multiple sensors within the city limits, thus their total number of observations during the KORUS-AQ timeframe is much larger than the other seven sites. Additionally, the HUFS site records data every minute while the others take hourly measurements. All four land classifications are represented within the group, although there is only one inland rural site. All

data during the KORUS-AQ timeframe are downloaded and the number of days with data ranges from 35-41. Overall, there are 51 monitors from Air Korea, six from the National Institute for Environmental Research (NIER), and one was from the Research Institute on Public Health and Environment (SIHE), and the last was from the Hankuk Institute of Foreign Studies (HUFS). It should be noted that 51 of the 59 data files had quality control techniques applied to them at the date of last access in August 2017

### 4 Methods and analysis

#### 4.1 Analysis of AOD Diurnal Variation

The AOD at each hour is computed by using instantaneous AERONET AOD measurements within ± 30 minutes centered over that hour. The hourly AOD for the same hour in different days is then averaged to compute the climatological (or baseline) AOD for that hour; the baseline AOD for each hour is then used to compute baseline daily-mean AOD, form which

the diurnal variation of AOD value for each hour can be subsequently calculated. This is completed for the hours of 0-10



UTC and 21-23 UTC due to the conversion to Korean Standard Time (KST). KST is nine hours ahead of UTC, so the diurnal variation period extends from 7-18 KST, as the first and last hour are excluded for a lack of observations.

Furthermore, to calculate the statistics for hourly variations, the percent difference from the (baseline) daily mean is computed. This is referred to as the percent departure from average. Our methods here are similar to the methods from Wang

et al. (2004) and Smirnov et al. (2002). Hence, expressing the departure as a percent allows for comparison to other AOD studies.

The original 22 AERONET sites are split into four land classifications to further analyze and define trends amongst their AOD diurnal variations at 550 nm. The sites of Chinhae, Korea University, and Kyungil Univsersity are excluded from the analysis due to the short data records, leaving 19 AERONET sites. Each site's full record of data is used in the calculation

(Table 1). Figure 1 shows a summary of this classification. Of the 19 sites, three are coastal urban, five are coastal rural, six are inland urban, and five are inland rural.

Shown in Fig. 2 is the AOD diurnal variation and percent departure from daily mean using the full record of data at all 19 AERONET sites, split into land classification. The coastal rural sites show the most similarity amongst each other and are characterized by AOD levels remaining virtually constant throughout the day as their departure from the daily mean is

generally ± 10%. This feature agrees with Smirnov et al. (2002) who also found that the Anmyon site, on the western coast of the Korean peninsula, has little to no diurnal variation of AOD at 500 nm when investigating a multiyear data record.

The AOD diurnal variation of the coastal urban sites is more pronounced with a departure from the daily average at ± 20% and has fewer similarities between sites versus the coastal rural classification. The KORUS NIER site has noticeably higher values of AOD than the Gangneung or Pusan for the majority of the day until 18 KST. These sites are urban in nature

and therefore have their own emissions characteristics, adding an additional layer of complexity to their distinctive diurnal variations. This could explain why the coastal urban sites experience more variation than the coastal rural sites. When focusing on the different diurnal variations amongst the coastal urban sites, we believe that the differences could stem from the site location. For example, the sites on the western Korean peninsula (i.e., KORUS NIER) are closer to Chinese emissions and may be impacted more by long-range transport than the coastal urban sites on the eastern Korean peninsula

(i.e., Pusan and Gangneung). When combined with local emissions, the external factors could lead to a different diurnal variation than their counterparts on the eastern side of the country.

All six of the inland urban sites show remarkably similar diurnal variations. Their AODs slightly decrease and remain constant until the early afternoon at approximately 14 KST when the AOD then gradually builds until 18 KST. One outlier to this evening-build characteristic is KORUS Olympic Park whose AOD values drop between 17 and 18 KST. As a whole,

their average departure from the daily mean is ±20% with the most negative values occurring in the midday (i.e, the inland urban sites experience a minima in AOD values during this time). The early morning and late evening increases could be attributed to an increase in traffic and transportation demands. It is interesting to note that this common diurnal variation trend is seen at sites that have as little as four months of data (i.e., KORUS Iksan) to greater than five years of data (i.e.,



Gwangju and Yonsei University). Below in section 4.3, we investigate this further to quantify how long of a record is needed at each site to match the diurnal variation produced by the full record.

The five inland rural sites naturally divide into two groups. KORUS Daegwallyeong, located on the eastern part of the peninsula, shows a much lower magnitude of AOD around 0.2-0.3 throughout the entire day. The other four locations of

Hankuk, KORUS Baeksa, KORUS Songchon, and KORUS Taehwa have higher AOD values near 0.4-0.6 in the morning before decreasing throughout the day and eventually rising again in the early evening at 15-16 KST. The only two exceptions to this trend are Hankuk and KORUS Songchon, located east-south-east of Seoul, that experience slight noontime increases. After the early evening build-up, most of the sites' AOD decreases at 17-18 KST, again with the exception of Hankuk and KORUS Songchon. The inland rural sites have the most variation for their percent departure from average with some sites

such as KORUS Baeksa and KORUS Songchon approaching -30% at 12 KST and 15 KST, respectively, and KORUS Daegwallyeong staying between ±10% with the except of 16 and 17 KST.

Overall, we see similar trends for the coastal sites and for the inland sites. The coastal urban and coastal rural sites have a lower average AOD value of 0.308 compared to the inland urban and inland rural sites whose average AOD value is 0.386. Additionally, regardless of land classification, most sites see an early morning and late afternoon maxima AOD and

noontime minima AOD. Factors influencing the diurnal variation include length of data record, number of available measurements for calculating the hourly averages, and site location compared to those sharing its land classification.

### 4.2 Analysis of Angström Exponent Diurnal Variation

Similar to the AOD diurnal variation, each site's full record of data is used to compute diurnal variation of the Angström Exponent. The same analysis procedure is used, but instead of calculating AOD at 550 nm, the only variable of interest is the

Angström Exponent between 440 nm and 675 nm. The Angström Exponent is of importance since it helps determine the aerosol's source. It is inversely related to the average size of aerosol particles, so the smaller the particles, the larger the value. Generally speaking, an Angström Exponent approaching 0 signifies coarse-mode or larger particles such as dust, and an Angström Exponent greater than or approaching 2 signifies fine-mode or smaller particles such as smoke from biomass burning (Wang et al. 2004). The size of the particle assists with attributing the aerosol to natural or anthropogenic sources

since the latter are typically smaller than their natural counterparts (of dust and sea salt particles).

All four land classifications show similar Angström Exponent values in the range of 1.2-1.6 with the exception of lower values near 1.0 at the inland rural sites (Fig. 3). All six of the inland urban sites experience a gradual build in Angström Exponent from 1.2 to 1.4 throughout the day. The coastal urban sites are similar, as Pusan and Gangneung also see an increase in Angström Exponent, but the KORUS NIER site has a noticeable Angström Exponent maxima near 1.2 at 10

KST. Also, its values are lower than the other coastal urban or inland urban sites, with a range of 1.1 to 1.2. However, the general trend of the coastal urban sites is that they too gradually increase in Angström Exponent as the day progresses.

Turning to the rural sites, all five of the inland rural sites experience the same gradual build throughout the day to 1.4 as the inland urban sites, but their values start a bit lower near 1.0 versus 1.2. The coastal rural sites hover ~1.3 for the majority




of the day, with KORUS UNIST Ulsan experiencing values as high as 1.5 from 12-16 KST before dropping back down to 1.4 by 17 KST. The Ångström Exponent diurnal variation of the coastal rural sites is not as pronounced as that in the other three land classifications that all experience morning Ångström Exponent minimums and gradually build throughout the day before plateauing.

With the typical range of 1.2-1.6 experienced at most sites, it is concluded that the majority of aerosols over the Korean peninsula are fine-mode particles with some coarser-mode particles seen overnight and in the early morning. As the day progresses, the particle size decreases due to secondary organic aerosol formation which leads to an increase in the Ångström Exponent.

**4.3 Observation Time for Climatologically-representative AOD Diurnal Variation**

In this section, we define the term "climatological diurnal variation". This is the diurnal variation pattern produced at each AERONET site in long term averages such that it is relatively persistent and statically robust. The concept is similar to the concept of climatology of diurnal variation of 2-m air temperature which, while varying with location, normally shows peak in the afternoon and minimum before the Sun rise [Wang and Christopher, 2006]. The concept of "climatological diurnal variation of AOD or aerosol properties", therefore builds upon the hypothesis that there are underlying processes inherent

with respect to a specific location to produce diurnal variation of AOD. For example, in the agricultural burning seasons over Central America, AOD values often peak around later afternoon and are minimal in the night before Sun rises; this is because such burnings often started in the late morning and diminishes at night [Wang et al., 2006]. Hence, an intriguing question is that how long our data record should be to obtain climatological diurnal variation of AOD or aerosol properties. We address this question by data collected at the five AERONET sites which have more than five years of data in their full

record: Anmyon, Baengnyeong, Gosan, Gwangju, and Yonsei University.

      To statistically compare how long of a data record is needed to match the climatological diurnal variation, we first compute the statistics starting from the first month of the data record to a certain number of months, *n*; hereafter the subset of the data is denoted as [1, n], with the first number being the starting of the month, and second number the last month in the subset. We then repeat the calculation by moving the starting month (and ending month) with increment of one month each

time, e.g., for the data subset [2, n+1], [3, n+2], …, [N-n+1, N], where N is the total number of months of whole data record. The average of diurnal variation statistics from each *n*-month data subset is then compared with statistics of diurnal variation from the whole data record. The comparison reveals the degree to which *n* month data record may describe the climatological diurnal variation derived from full data record for a specific site of interest. We then repeat the same process by increasing the number of months for the subset (n) by one month, two months, three months, …, until the subset

eventually grows to the full record. It is expected that as *n* increases, the climatological diurnal variation will be better characterized.



The actual implementation of the method above requires the removal of the gaps of missing data, e.g., months that don't have observation. Hence, Anmyon, Baengnyeong, Gosan, Gwangju, and Yonsei University are left with full records of 89, 53, 80, 82, and 71 months, respectively, for the analysis.

Figure 4 shows that as $n$, the number of months of observation increases, the diurnal variation being described by these subset observation is in more agreement with the counterpart from the full record, in terms of linear correlation coefficient R, root mean square error (RMSE), and statistical significance. The inland urban sites of Gwangju and Yonsei University have very similar results (Figs. 4d and 4e). Gwangju requires 13 months of data to become significant ($p < 0.05$), or roughly 15.9% of the full (82 months) record of data. Yonsei requires 11 months of data which is slightly lower at 15.5% (of 71 months). Additionally, they both have a R-value around 0.8 at the occurrence of the first significance, and their results become significant shortly after the RMSE and R-value graphs intersect. We conclude that the inland urban sites require 10-12 months of data to match the climatological diurnal variation with an R value of 0.8 and $p < 0.05$. Additionally, they require 45-47 months of data for an RMSE < 0.02.

The results for coastal rural sites of Anmyon, Baengnyong, and Gosan are less c (Fig. 4 a-c). They require more data than the urban sites before becoming significant ($p < 0.05$) with a range from 18% to 32%. Anmyon requires 18% of the total data with 16/89 months and Baengnyeong and Gosan require about 30% of the total data with 17/53 and 24/80 months, respectively. They have slightly lower R-values of 0.57-0.78 at the time of their first significance, and similar to the inland urban sites, their correlations become significant after the intersection of the RMSE and R-value graphs. These three sites require 21-25 months of data to match their climatological diurnal variations with an R value of 0.8 and $p < 0.05$, which is twice of that of the inland urban sites. Overall, RMSE < 0.02 can be reached with 35-52 months of data in the subset; this is similar to the inland urban sites, but a much greater range. The varied location of these three coastal rural sites could explain the variability between them, as they extend from Baengnyeong near North Korea to Gosan off the southern coast of the Korean peninsula.

Between the five sites that we investigated, there is no consistent pattern among the number of months of full record data required for diurnal variation replicability and significance. Baengnyeong, the site with the least amount of full record data (53 months), requires 16 months (32%) of the data, while Yonsei (71 months of data) requires 11 months (15.5%), Gosan 24 months, Gwangju 13 month, Anmyon 16 months. In average, ~18 months of observation data are needed to obtain statistically significant results for characterizing the diurnal variation of AOD.

One findings surprising but interesting to note is that coastal rural sites would require twice longer observation than the inland urban sites to match the climatological diurnal variation with R = 0.8 and $p < 0.05$. We thought that due to the complexity of the urban sites having both their own emissions and those via background and transport, they would require more data for a common trend to emerge. However, it is in fact just the opposite, suggesting that diurnal variation of AOD in urban setting is distinct and persistent.



### 4.4 Analysis of WRF-Chem and GOCI AOD

The WRF-Chem model provided hourly chemical weather forecasts from 1 May 2016 to 15 June 2016. Thus, this is defined as the KORUS-AQ timeframe, and spatial and temporally matched data pairs between GOCI-AEROENT and WRF-Chem AERONET are analyzed. Because of clouds, GOCI AOD data is rarely eight time per day, while the AERONET data also
undergoes its own quality control algorithms. In contrast, the WRF-Chem data is available for every hour during the timeframe of interest. Due to these factors, the intercomparision dataset is the smallest between AERONET vs GOCI (1,583 data pairs) and the largest between AERONET vs WRF-Chem (3,633 data pairs).

    The scatter plots between observation (x-axis) and either model or GOCI (y-axis) is shown in Fig. 5, while the color of each scatter plot point represents the density of observations within that data range. Although both are statistically significant
($p < 0.001$), the AERONET vs GOCI R-value of 0.8 is double that of the AERONET vs WRF-Chem R-value of 0.4. Additionally, the RMSE value between AERONET vs GOCI is lower as well. Overall, WRF-Chem can both over- and under-predicted the AERONET AOD values while the GOCI satellite typically underpredicted them during the KORUS-AQ timeframe. In reference of the AERONET AOD values, GOCI AOD is better than WRF-Chem predictions.

    To analyze these results on a site basis, Taylor Diagrams are produced (Fig. 6), in which the radius represents the
normalized standard deviation while the correlation coefficient is represented as cosine of polar angle. Hence, regardless of the data sources, the observation data will be represented at the one point (with normalized standard deviation of 1 and cosine of polar angle of 1). The color of data points shows the data bias with respect to the observation.

    Figure 6a shows that WRF-Chem both over- and under-predicted AOD values during the campaign, and there is no clear indication whether the WRF-Chem performed consistently better for certain land classification (within 15 sites we analyzed).
For example, looking at the inland urban sites, WRF-Chem underpredicted two sites (KORUS Olympic Park and KORUS Iksan), overpredicted two sites (Gwangju and KORUS Kyungpook), and had no bias at one site (Yonsei University). The R values range from 0.15 to 0.7 with the majority of points between 0.15 and 0.4. Lastly, there is an even distribution of sites where the WRF-Chem standard deviation is higher than and less than the AERONET standard deviation.

    Figure 6b shows the relationship between AERONET and GOCI at the same 15 sites during the KORUS-AQ campaign.
Here, we see that only three sites (Baengnyeong, KORUS NIER, and KORUS Daegwallyeong) had a positive bias and thus were over predicted by GOCI. Noticeably different from 6a is the shifted R value range, now extending from 0.35 to 0.9 but concentrated within 0.7 to 0.9. Another interesting difference is that the GOCI standard deviation was greater than the AERONET standard deviation at all 15 sites, suggesting COCI AOD tends to amply the temporal variation of AOD.

### 4.5 PM$_{2.5}$ Diurnal Variation

Table 2 lists information for all PM$_{2.5}$ sites, including their full record of data, the number of recorded observations within that full record, the nearby AERONET station, the number of days having data recorded, and the hours of minima and maxima PM$_{2.5.}$ Nine of the ten sites reported data in hourly averages. For the HUFS site whose data was reported minutely,





the hourly averages are created by computing the mean of data in that hour, as is done by most analysis of hourly $PM_{2.5}$. Again, only 7-18 KST is analyzed for $PM_{2.5}$.

As seen in Fig. 7, the $PM_{2.5}$ diurnal variations are surprisingly invariant given the short timeframe of interest. One would expect that as the data timeframe decreases, the fluctuations would increase. However, this is not the case for $PM_{2.5}$. The

coastal urban sites of Busan and Bulkwang remain near 30-35 µg m$^{-3}$ throughout the day (Fig. 7a). Bulkwang does slightly increase at 8 KST but then returns to the constant value of 30 µg m$^{-3}$ by 11 KST. The inland urban sites (Fig. 7b) closely resemble the coastal urban sites in diurnal pattern and value except for Olympic Park. Its values were double the other inland and coastal urban sites at 60 µg m$^{-3}$ at its 8 KST maximum. Concentrations then drop to near 45 µg m$^{-3}$ for 13-18 KST. Olympic Park shows the most fluctuations throughout the day in its $PM_{2.5}$ concentrations.

The coastal rural sites (Fig. 7c) of Jeju, Baengnyeong, and KORUS UNIST Ulsan split into three diurnal variations that are varied in concentration and each with its own unique pattern. The KORUS UNIST Ulsan site has the highest concentrations near 30 µg m$^{-3}$ but exhibits little to no fluctuations throughout the day. The Baengnyeong site has the second highest concentrations near 25 µg m$^{-3}$ with a 14 KST maximum. Jeju has the lowest concentrations near 15 µg m$^{-3}$ and maxima at 11 and 14 KST with a 10 and 13 KST minima. HUFS, the inland rural site (Fig. 7d), fluctuates between its 8 KST

maxima near 40 µg m$^{-3}$ and drops to a concentration near 30 µg m$^{-3}$ by 12 KST and remains there for the rest of the day.

Overall, seven of the ten sites have stagnant $PM_{2.5}$ values near 30 µg m$^{-3}$ throughout the entire day. The other three sites, Bulkwang, Daejeon, and HUFS all have morning $PM_{2.5}$ maxima at 8 KST before dropping to stagnant values by 13 KST. Daejeon experiences abnormally highest $PM_{2.5}$ concentrations with values peaking near 60 µg m$^{-3}$. For all sites, the correlation between hourly $PM_{2.5}$ variation and daily-mean $PM_{2.5}$ variation (Fig. 8) has the highest $R^2$ value above 0.8 at

noontime, and decreases toward early morning and later afternoon, reaches the minimum at mid-night, which suggests that day-time variation of emission and boundary layer process are dominant factors affecting day-to-day variability of $PM_{2.5}$.

**4.6 AOD-$PM_{2.5}$ Diurnal Variation Relationship**

Here we study which data source (from GOCI or WRF-Chem) either predicted or retrieved the AERONET AOD values better. We also compare the AOD diurnal variation to the $PM_{2.5}$ diurnal variation. In Fig. 9, the diurnal variations are shown

for AERONET, WRF-Chem, GOCI, observed $PM_{2.5}$, and WRF-Chem predicted $PM_{2.5}$ for the sites that have all five dataset available (ie: the sites listed in Table 2). All data is temporally and spatially matched. Due to its retrieval times, GOCI's AOD diurnal variation only extends from 9-16 KST, thus only 9-16 KST is used for AERONET and WRF-Chem as well.

The GOCI AOD values better matched the observed AERONET AOD diurnal variation. As seen in Fig. 9a, although hourly GOCI AOD has a systematic low bias of 0.02-0.05 with respect to the AEROENT counterparts, the GOCI AOD

diurnal variation (green line) mirrors that of AERONET (blue line) for the entire day, showing low values around noon and dual peaks (one in 10-11 KST) in the morning and (another 14 KST) in the afternoon, respectively; both GOCI and AERONET also shows that the minimum AOD is in the later afternoon at 16 KST, although GOCI shows a relatively larger





decrease of AOD from 14 KST to 16 KST. In contrast, while WRF-Chem AOD values are consistent with GOCI and AOERNET to describe the dual peaks and low AOD values around noon, a much stronger peak at 16 KST (than that at 10 KST) in WRF-Chem differs that from GOCI and AEROENT (both of which show comparable dual peaks). Furthermore, WRF-Chem shows a relatively increase of (minimum) AOD at 9 KST to 15 and 16 KST, while GOCI and AEROENT both

show the decrease with minimum AOD at 16 KST. Hence, it is hypothesized that the diurnal emission in WRF-Chem may have too much skewness toward afternoon emission; indeed WRF-Chem AOD is comparable to AERONET AOD values in the morning, but shows large positive bias up to 0.08 in the later afternoon. This hypothesis needs to be further studied.

Also plotted in Fig. 9a is the average observed and WRF-Chem predicted $PM_{2.5}$ diurnal variation of the 10 $PM_{2.5}$ sites. The observed concentrations peak at 10 KST with a value approaching 33 $\mu g\ m^{-3}$ but drop to less than 28 $\mu g\ m^{-3}$ by 13 KST,

and then peak again to 30 $\mu g\ m^{-3}$ at 14 KST. WRF-Chem predicted $PM_{2.5}$ is systematically higher than observed $PM_{2.5}$ by 10-15 $\mu g\ m^{-3}$, but it has similar dual peaks at 10 and 14 KST. Similar as its AOD variables, WRF Chem showed that the peak at 14 KST is ~3 $\mu g\ m^{-3}$ higher than the peak at 10 KST, while observed $PM_{2.5}$ show that the peak at 14 KST is ~5 $\mu g\ m^{-3}$ lower than the peak at 10 KST. Furthermore, WRF-Chem shows an increase of $PM_{2.5}$ from 11 to 13 KST while the observed $PM_{2.5}$ showed the opposite. Overall, the observed $PM_{2.5}$ concentrations decrease from morning (9-10 KST) to

evening (15-16 KST), but WRF-Chem counterparts show the opposite. Hence, the comparison and contrast analyses of both AOD and $PM_{2.5}$ suggest further studies for the diurnal variation of emission in WRF-Chem.

In general, the diurnal variation of AERONET AOD (*averaged over all sites*) fluctuates the least throughout the day with a percent departure from the daily mean of ± 6%. WRF-Chem fluctuates ± 8% while GOCI shows the most variation at +9% to -30% due to an outlying low value at 16 KST. Similar to AERONET and WRF-Chem, the $PM_{2.5}$ percent departure from

daily mean ranges from ± 8%.

Figure 9b displays the diurnal variation of the $PM_{2.5}$/AOD ratio derived from WRF-Chem and collocated AERONET and $PM_{2.5}$ measurement throughout the KORUS-AQ campaign. This ratio is valuable because satellite AOD often is used to multiply this ratio to derive surface $PM_{2.5}$. While the ratio from WRF-Chem overall is larger than observation-based counterparts by 30-50% in all sites (except Daejeon for some hours and one outliner at a particular hour in Gwangju site), the

majority of the ratios range from 60-140 $\mu g\ m^{-3}\ \tau^{-1}$ with outliers as low as 40 and as high as 160 $\mu g\ m^{-3}\ \tau^{-1}$. Overall, there is no apparent trend between $PM_{2.5}$/AOD ratio and time of day. This conclusion is consistent even when analyzing based on land classification. The three coastal rural sites of Baengnyeong, Jeju, and Ulsan have ratio maximums in both the morning (7, 8, and 10 KST) and early evening (17 KST). Their minimums range from morning (9 KST), noontime, and late afternoon (16 KST). The two coastal urban sites of Bulkwang and Busan show more similarities with peaks in the early morning (8 and

9 KST) but still have a minimum range from noontime to afternoon (12 and 15 KST). The inland urban sites have morning (9 and 10 KST), afternoon (13 KST), and early evening (17 and 18 KST) maximums but cohesively have a 15 KST minima, aside from Gwangju whose ratio steadily increases after 10 KST. Being the only inland rural site, the $PM_{2.5}$/AOD at HUFS



has a maxima at 8 KST and steadily decreases afterward for the remainder of the day. This conclusion suggests that diurnal variation is not a prominent factor in using the PM$_{2.5}$/AOD ratio to derive PM$_{2.5}$ values from AOD.

## 4 Summary and conclusions

By using all possible AEROENET data in South Korea, the surface observation of PM$_{2.5}$, GOCI AOD and WRF-Chem
simulated AOD during KORUS-AQ Field Campaign in South Korea from April to June 2016, this study analyzed the diurnal variation of aerosol properties and surface PM$_{2.5}$ from surface observations, and assessed their counterparts from models. In summary, the following were found.

1.  Long-term AEROENT data shows that the climatological AOD diurnal variation is very similar amongst South Korean AERONET sites. Most see an AOD maxima in the middle morning (10 am) and middle afternoon (2 pm)
and a noontime AOD minima. Additionally, the coastal sites have lower average values near 0.3 at 550 nm while the inland sites have higher values near 0.4. The inland sites also experience the most AOD fluctuations during the day on the order of +20% to -30%. Analysis of the Angström Exponent shows a gradual increase throughout the day from 1.2 to 1.4.

2.  Given there is a persistent diurnal variation of AOD and Angstrom Exponent in South Korea, we analyzed that at
minimum, there should be more than 12 months of observation, and the coastal rural sites require twice of observations than the inland urban sites, to characterize the climatology of diurnal variation of AOD at statistically significant level. This suggests the distinct and persistent diurnal variation of aerosol properties in urban areas.

3.  The AERONET and GOCI AOD had a linear correlation coefficient of (R) 0.8 and RMSE = 0.16 while the AERONET and WRF-Chem relationship had R = 0.4 and RMSE = 0.28, suggesting that AOD data retrieved from
GOCI satellite shows a closer agreement with AERONET AOD data than those from WRF-Chem model.

4.  Analysis of 10 AERONET-surface PM$_{2.5}$ paired sites show that the diurnal variation of PM$_{2.5}$ was ~10% throughout the day, with the exception of the Daejeon and HUFS sites having a maxima at 8 KST (or peaks by 20%) and values gradually decreasing and remaining steady for the remainder of the day after 12 KST. PM$_{2.5}$ daily-mean values were around 30 μg m$^{-3}$ which is still 20 μg m$^{-3}$ below the 24-hour PM$_{2.5}$ air quality standard in South Korea but 5 μg m$^{-3}$
above the WHO recommendation. Overall, the day-to-day variation of mean PM$_{2.5}$ at all sites can be best described by the variation of hourly PM$_{2.5}$ data at noontime for each day, and is least captured by the variation of PM$_{2.5}$ in the mid-night hours.

5.  AERONET, GOCI, WRF-Chem, and observed PM$_{2.5}$ data consistently show dual peaks for both AOD and PM$_{2.5}$, one at 10 KST and another that 14 KST. However, WRF-Chem show the peak in afternoon is larger than the peak in
the morning, which is opposite from what GOCI and AERONET reveal. Consequently, WRF-Chem shows increase of AOD and PM$_{2.5}$ from 9 KST to 16 KST, which contrasts with the deceasing counterparts in GOCI, AEROENT,



and observed $PM_{2.5}$. The analysis suggests that the diurnal profile of emissions in WRF-Chem may have a too larger skewness toward the afternoon.

6. $PM_{2.5}$/AOD ratio ranged from 60-120 throughout the day, and no consistent pattern was seen at the 10 sites nor when further broken down into land classification. The ratio in WRF-Chem is persistently larger than the observed

counterparts by 30-50% in different sites. This highlights the combined need to use satellite to characterize aerosol 2D information and use chemistry transport models to resolve the space of vertical prolife toward improved estimate of surface $PM_{2.5}$.

By using rich data sets during KORUS-AQ, this study revealed there are persistent diurnal variation of AOD and surface $PM_{2.5}$ in South Korea. It is shown that the Korean GOCI satellite is able to consistently capture the diurnal

variation of AOD, while WRF-Chem clearly has the deficiency to describe the relatively change in the morning and afternoon. As a minimum of one-year observation is required to fully characterize the climatology of diurnal variation pattern of AOD, future field campaigns are commended to have at least longer time periods of surface observations where AEROENT and surface $PM_{2.5}$ network can be collocated. Hence, future studies are needed to evaluate the statistical significance of our analysis of diurnal variation of $PM_{2.5}$/AOD ratios with a longer record of observation data.

**Acknowledgement**

This research is in part supported by NASA's GEO-CAPE program and in part by KORUS-AQ program (grant #: NNX16AT82G). AERONET data are downloaded from http://aeronet.gsfc. nasa.gov and we thank all AERONET PIs in South Korea for collecting the data.

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



## Tables

Table 1. Details for all 22 AERONET sites. $N_{Full}$ and NKORUS denote the number of observations. $N_{Days}$ denotes the number of days with observations in the full record of data. The hour(s) of AOD minimum/maximum correspond with Fig. 2.

| Land Classification/Site | Full Record | $N_{Full}$ | $N_{DAYS}$ | KORUS-AQ Record | $N_{KORUS}$ | Hour(s) of AOD Minimum (KST) | Hours(s) of AOD Maximum (KST) |
|---|---|---|---|---|---|---|---|
| Coastal, Rural | | | | | | | |
| Anmyon | 10/17/99 - 12/09/15 | 26,147 | 1,351 | -- | -- | 9 | 18 |
| Baengnyeong | 07/25/10 - 08/31/16 | 21,113 | 844 | 05/01/16 - 6/13/16 | 1,949 | 16 | 7, 18 |
| Gosan | 04/04/01 - 09/12/16 | 24,208 | 982 | 05/01/16 - 6/14/16 | 1,506 | 10 | 16 |
| KORUS Mokpo | 03/02/16 - 01/08/17 | 7,602 | 196 | 05/01/16 - 6/14/16 | 1,728 | 14 | 7, 17 |
| KORUS UNIST Ulsan | 03/03/16 - 02/01/17 | 11,441 | 231 | 05/01/16 - 6/11/16 | 1,989 | 12 | 7, 18 |
| Coastal, Urban | | | | | | | |
| Chinhae | 04/21/99 - 09/15/99 | 796 | 54 | | | -- | -- |
| Gangneung | 06/03/12 - 12/16/15 | 13,272 | 569 | -- | -- | 11 | 7, 18 |
| KORUS NIER | 02/29/16 - 06/10/16 | 1,857 | 80 | 05/01/16 - 6/10/16 | 940 | 9, 18 | 8, 13 |
| Pusan | 06/16/12 - 02/11/17 | 16,253 | 536 | 05/01/16 - 6/11/16 | 2,331 | 9, 14 | 7, 18 |
| Inland, Rural | | | | | | | |
| Hankuk | 06/01/12 - 12/21/16 | 17,341 | 603 | -- | -- | 14 | 7, 18 |
| KORUS Baeksa | 04/24/16 - 06/14/16 | 2,536 | 46 | 05/01/16 - 6/14/16 | 2,190 | 12 | 7, 17 |
| KORUS Daegwallyeong | 03/10/16 - 06/14/16 | 3,152 | 75 | 05/01/16 - 6/14/16 | 2,416 | 13 | 10, 16 |
| KORUS Songchon | 04/22/16 - 06/13/16 | 2,576 | 47 | 05/01/16 - 6/13/16 | 2,208 | 15 | 9, 18 |
| Kyungil University | 11/12/12 - 12/12/12 | 617 | 28 | | -- | -- | -- |
| KORUS Taehwa | 04/11/16 - 06/13/16 | 1,939 | 50 | 05/01/16 - 6/13/16 | 1,712 | 12, 14 | 8, 17 |
| Inland, Urban | | | | | | | |
| Gwangju | 01/04/07 - 05/20/16 | 25,531 | 1,368 | 05/03/16 - 5/20/16 | 3,980 | 14 | 7, 18 |
| Korea University | 06/01/12 - 07/26/12 | 1,407 | 34 | | -- | -- | -- |
| KORUS lksan | 03/03/16 - 06/10/16 | 2,725 | 83 | 05/01/16 - 6/10/16 | 1,714 | 12 | 7, 17 |
| KORUS Kyungpook | 03/02/16 - 02/14/17 | 10,443 | 238 | 05/01/16 - 6/14/16 | 2,399 | 13 | 7, 18 |
| KORUS Olympic Park | 05/01/16 - 06/14/16 | 2,192 | 39 | 05/01/16 - 6/14/16 | 2,192 | 14, 18 | 7, 17 |
| Seoul | 02/15/12 - 07/30/15 | 8,175 | 382 | | -- | 14 | 7, 18 |
| Yonsei University | 03/04/11 - 01/16/17 | 65,277 | 1,542 | 05/01/16 - 6/13/16 | 2,271 | 10 | 7, 18 |




Table 2. The 10 KORUS-AQ ground sites with PM$_{2.5}$ data and their corresponding AERONET station. * means that only data from 5/1/16 – 6/10/16 is used. ** means that the data are minutely versus the others that are hourly, hence the large number of observations. *** similarly means that only data from 5/9/16 – 6/15/16 is used since the KORUS-AQ timeframe is defined in this study as 5/1/16 – 6/15/16. Sites with a number in parenthesis denotes how many individual stations were within that city, contributing to the higher number of observations.

| Land Classification /PM$_{2.5}$ Site | Full Record | N$_{Full}$ | AERONET Site | N$_{DAS}$ | Hour(s) of PM2.5 Minimum (KST) | Hours(s) of PM$_{2.5}$ Maximum (KST) |
|---|---|---|---|---|---|---|
| **Coastal, Rural** | | | | | | |
| Baengnyeong | 05/08/16 - 06/12/16 | 825 | Baengnyeong | 36 | 7, 18 | 14 |
| Jeju | 05/08/16 - 06/12/16 | 632 | Gosan | 35 | 10, 13 | 11, 14 |
| Ulsan | 05/09/16 - 06/12/16 | 840 | KORUS UNIST Ulsan | 35 | 11 | 18 |
| **Coastal, Urban** | | | | | | |
| Bulkwang | 05/08/16 - 06/12/16 | 839 | KORUS NIER | 36 | 13 | 8 |
| Busan (19) | 04/29/16 - 06/10/16* | 19,192 | Pusan | 41 | 10 | 17 |
| **Inland, Rural** | | | | | | |
| HUFS | 05/08/16 - 06/12/16 | 48,183** | KORUS Songchon | 36 | 18 | 8 |
| **Inland, Urban** | | | | | | |
| Daejeon | 05/08/16 - 06/12/16 | 841 | KORUS Iksan | 36 | 15 | 8 |
| Gwangju (7) | 04/29/16 - 06/10/16* | 6,794 | Gwangju | 41 | 7 | 17 |
| Olympic Park | 05/09/16 - 06/17/16*** | 917 | KORUS Olympic Park | 38 | 15, 18 | 8, 17 |
| Seoul (26) | 04/29/16- 06/10/16* | 26,095 | Yonsei University | 41 | 16 | 10 |



**Figures**

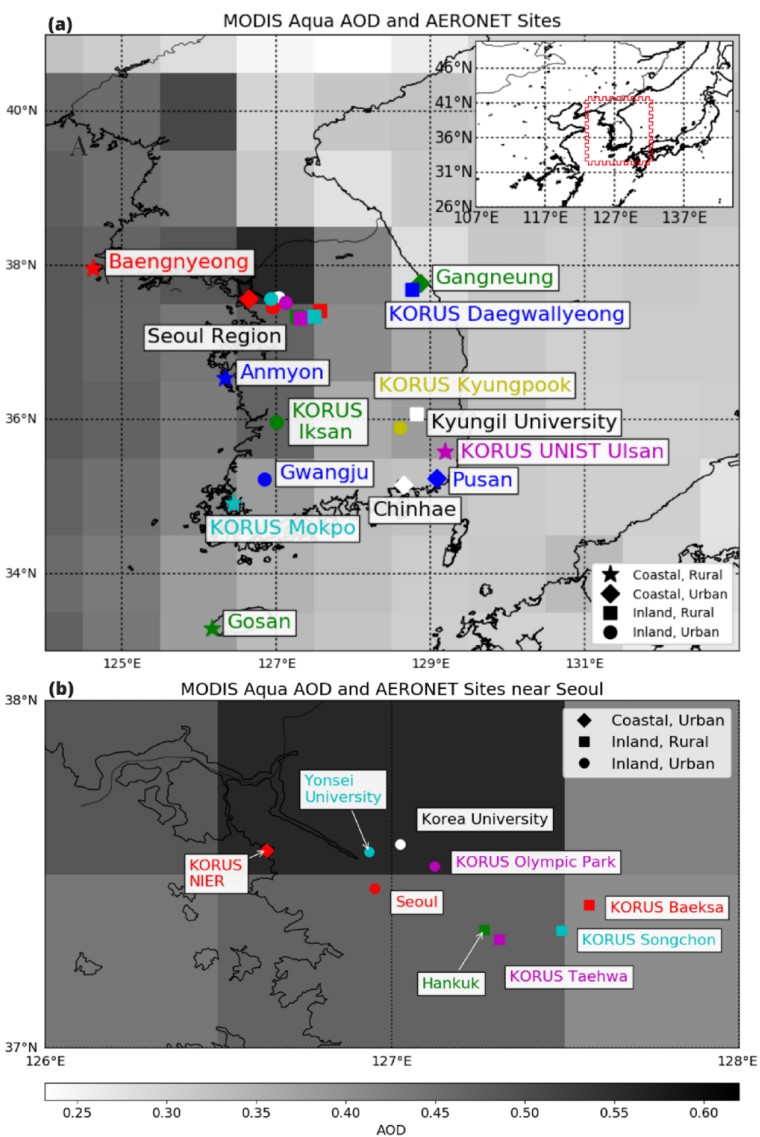

**Figure 1. a) Map of AERONET sites used in this study. The site marker corresponds to its land classification and its color corresponds to Figure 2. Sites with white markers are not used in Figure 2 due to their limited data availability. Overlaid is AOD 550 nm Dark Target from MODIS Aqua. The daily mean data are used. b) Zoom in of the Seoul Region.**





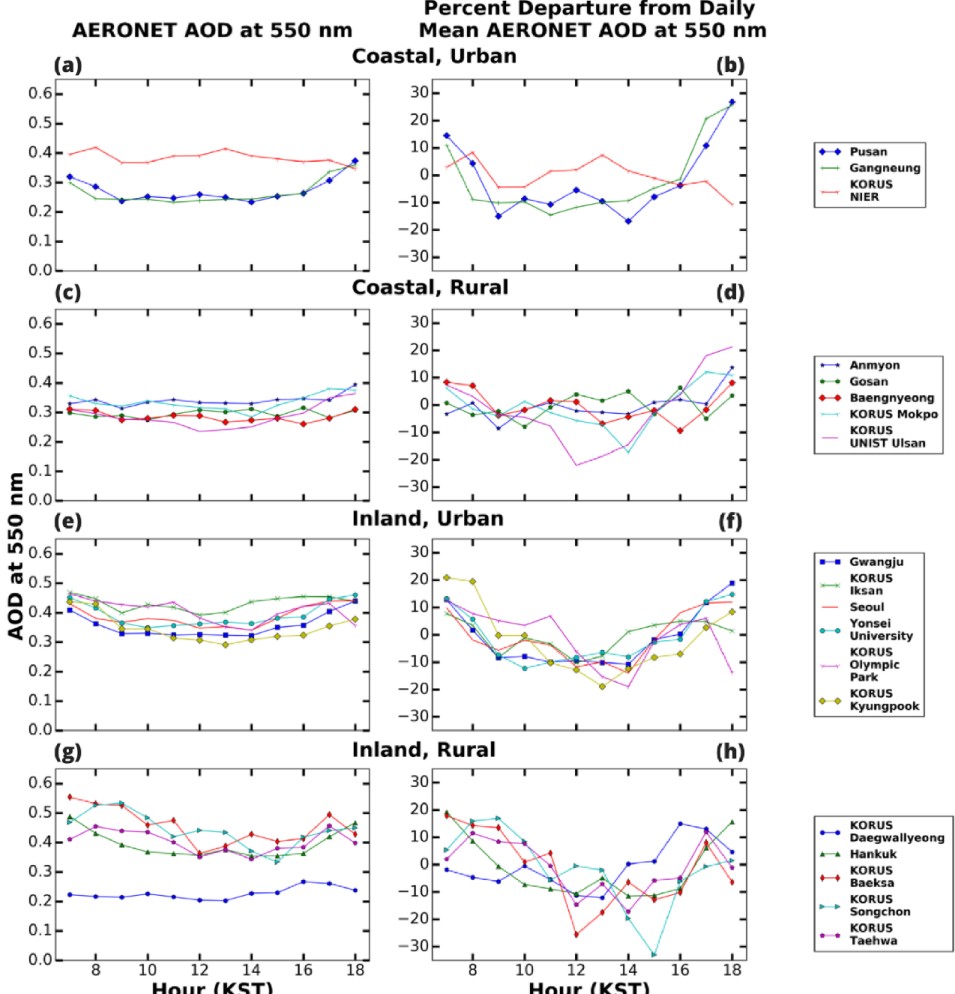

**Figure 2.** The AOD diurnal variation and percent departure from daily mean using the full record of AERONET AOD at 550 nm data for the 19 sites of interest.



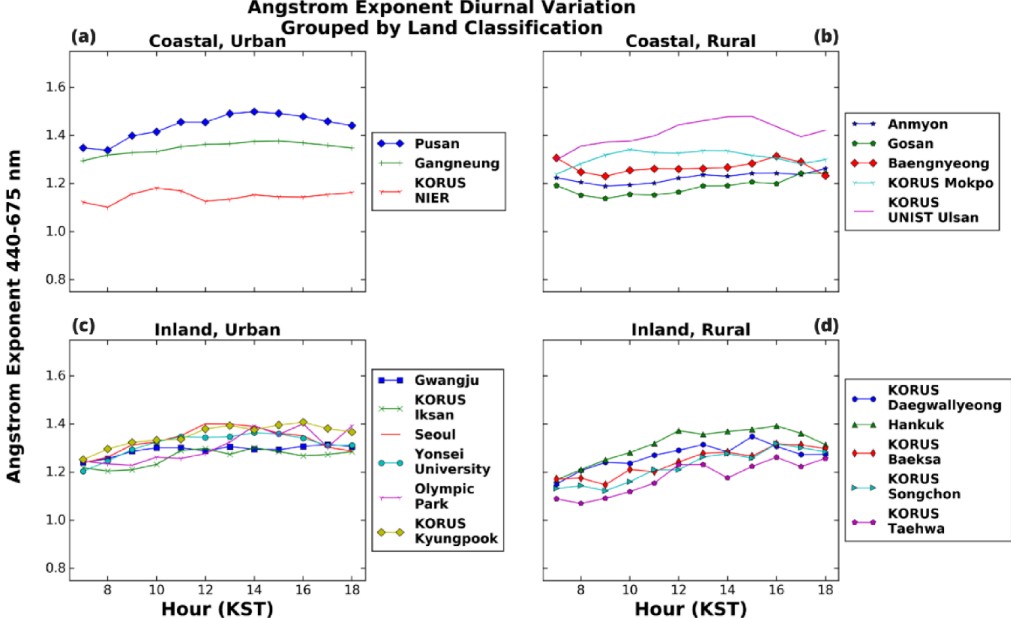

**Figure 3. The Angstrom Exponent 440-675 nm diurnal variation using the full record of AERONET data for the 19 sites of interest.**





**Figure 4. Establishing how long of a record of data is needed to match the "climatological" AOD at 550 nm diurnal variation. Anmyon, Baengnyeong, and Gosan are all coastal, rural sites. Gwangju and Yonsei University are inland, urban sites. The R value is in blue, the RMSE value is in red, and the p-value corresponds to the marker characteristic. A filled in marker represents p < 0.05 and an open marker represents p > 0.05.**



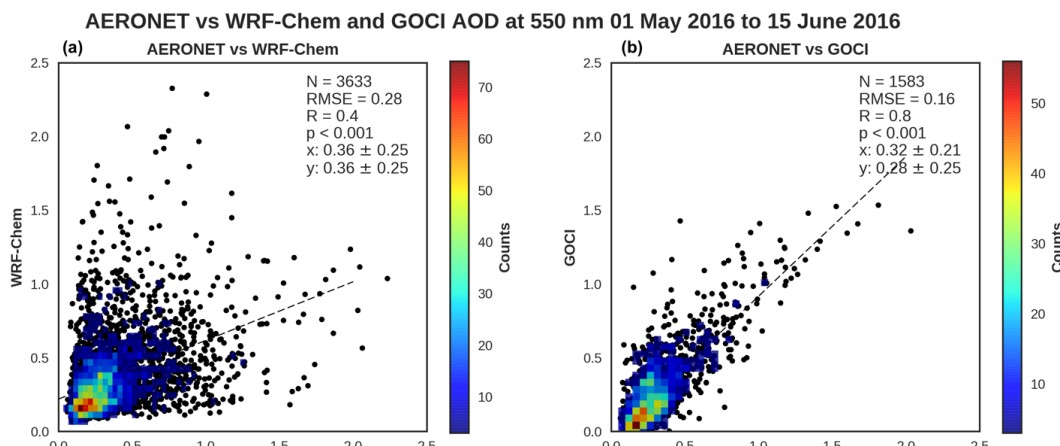

**Figure 5. AERONET vs WRF-Chem and AERONET vs GOCI AOD at 550 nm for the KORUS-AQ Campaign from      01 May 2016 to 15 June 2016. Only the temporally and spatially matched data are used. The point color corresponds to the density of observations within that area. Note that although the same color, the color bar ranges are different.**



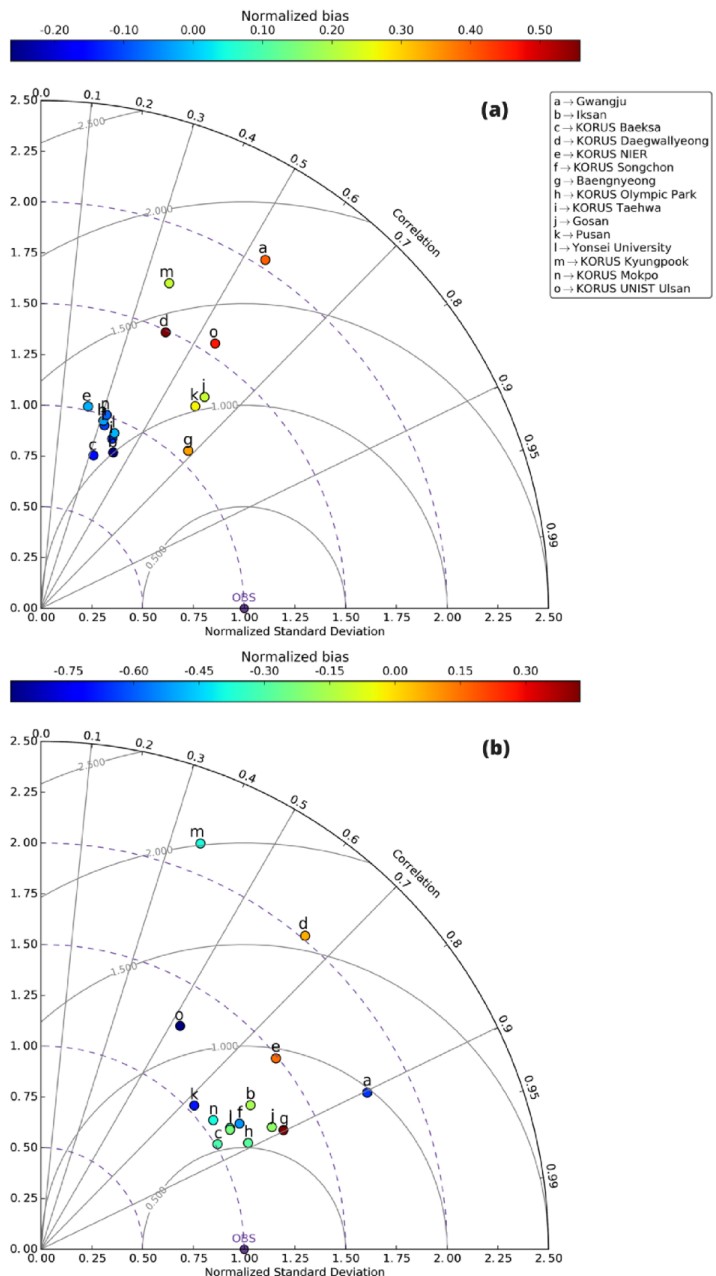

**Figure 6. Taylor Diagrams of a) AERONET vs WRF-Chem and b) AERONET vs GOCI**



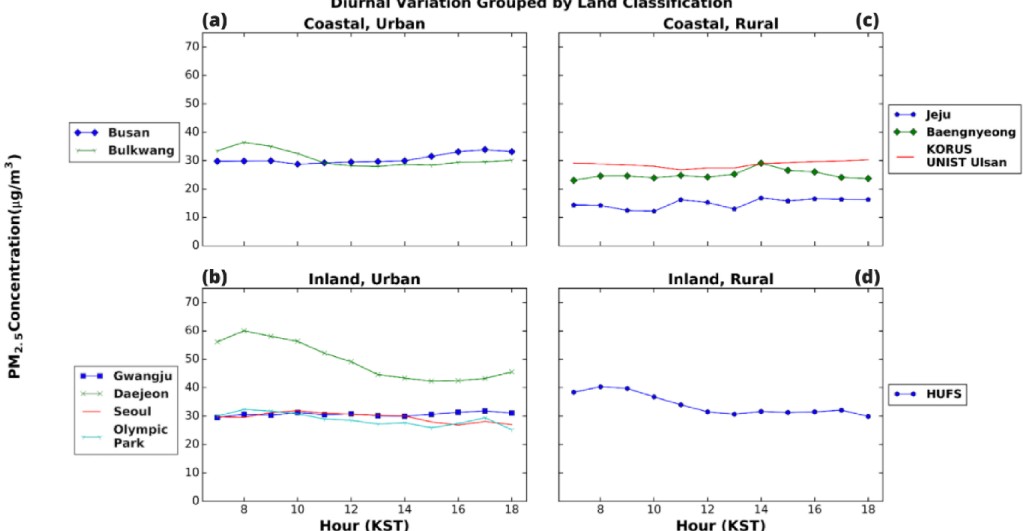

**Figure 7. The PM2.5 diurnal variation using the 10 KORUS-AQ ground sites that have a corresponding AERONET station nearby. The 24 hour PM2.5 air quality standard in South Korea is 50 mg m-3 and the WHO recommendation is 25 mg m-3.**

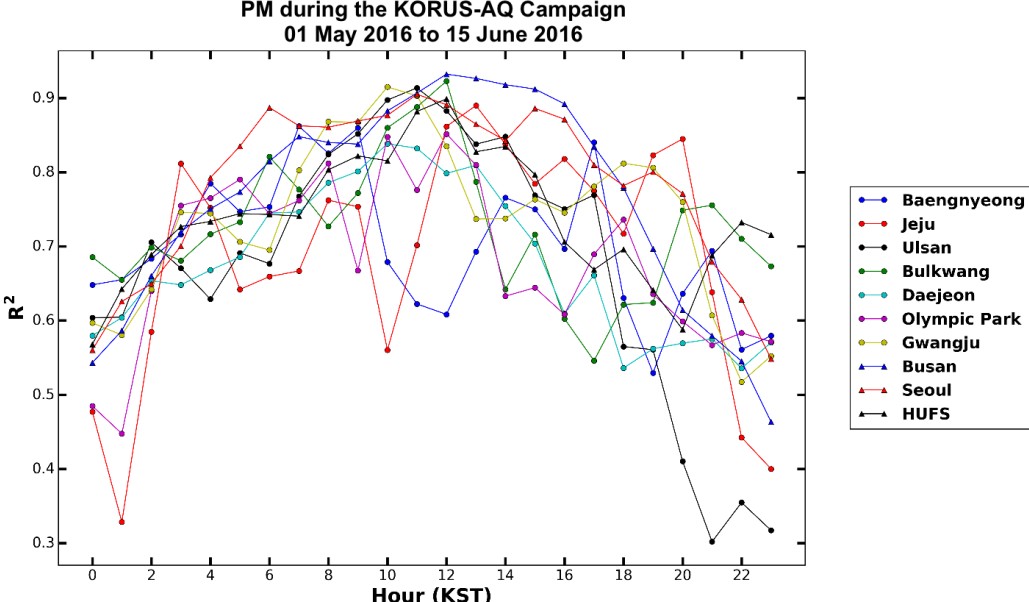

5    **Figure 8. The diurnal variation of R2 for PM2.5 for the sites in Table 2.**





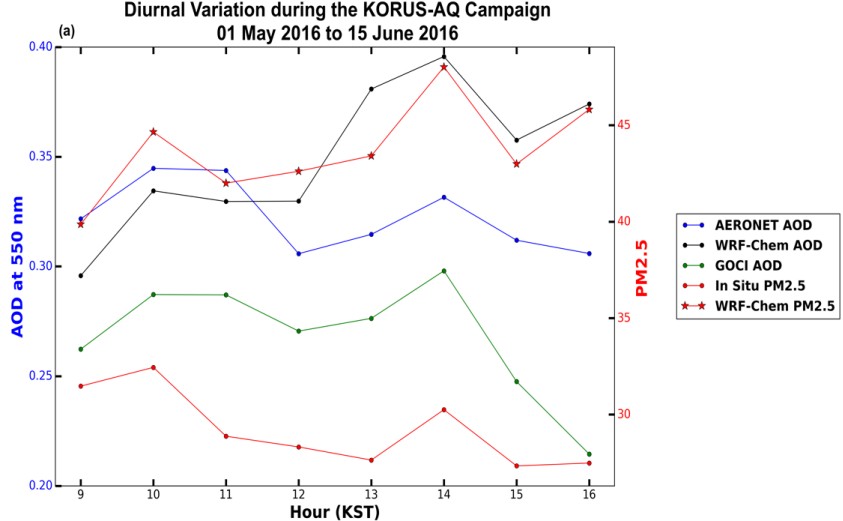

**Figure 9. (a)** The diurnal variations for AERONET, WRF-Chem, GOCI, and PM2.5 for the sites in Table 2 using temporally and spatially matched data. **(b)** The diurnal variation of the observation and WRF-Chem PM2.5 /AOD ratio for the average of the sites in Table 2 using temporally and spatially matched data.





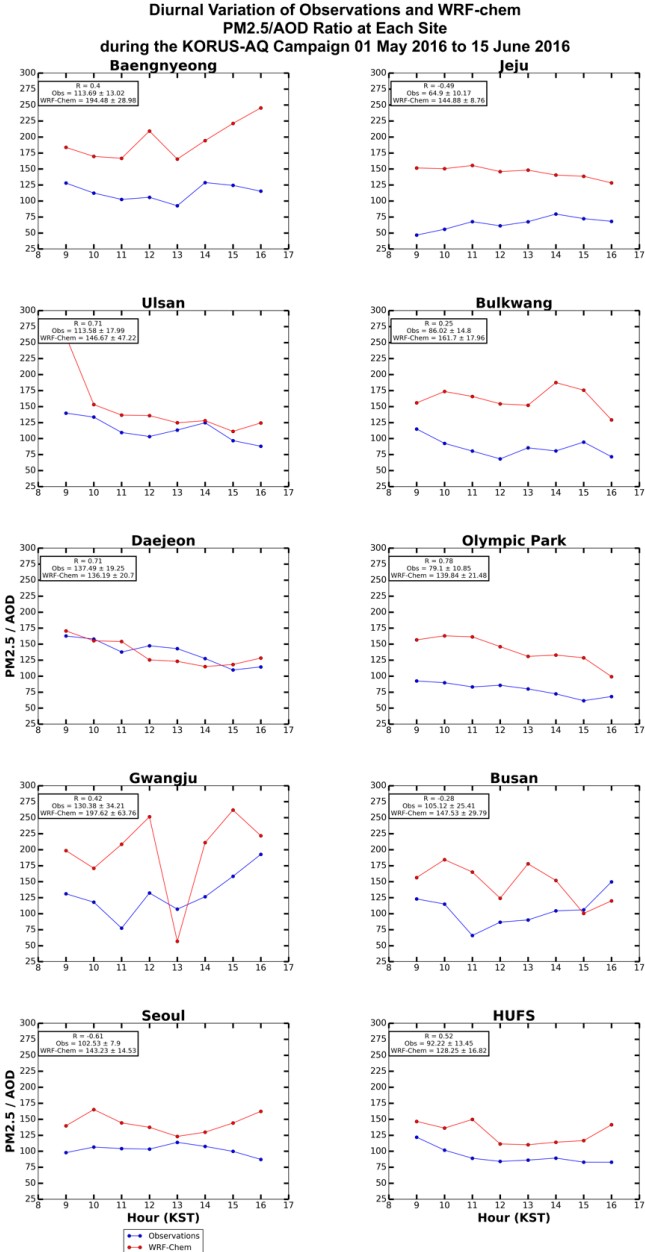

**Figure 10. The diurnal variation of the observed and WRF-Chem PM2.5 /AOD ratio for each site in Table 2 using temporally and spatially matched data.**