# Peer review of "Diurnal variation of aerosol optical depth and PM2.5 in South Korea: a synthesis from AERONET, satellite (GOCI), KORUS-AQ observation, and WRF-Chem model"

_Atmospheric Chemistry and Physics, 2018_

## Referee Comment (RC1) · C. Sioris (Referee) · 30 May 2018

Review of Lennartson et al. (submitted to ACP)

Overall comments:

I recommend this paper for publication after some suggested revisions. This is an interesting study that explores different dimensions (spatial variations, diurnal variations, etc.).

The paper clearly needs to be improved in terms of use of English. Some corrections that I suggested at quick-review stage were not included in the current manuscript. The corrections or additions that were already made to the manuscript will be helpful for the readers.

GOCI does not have a channel longer than 865 nm. This seems like a weakness since it must be difficult to retrieve an unbiased estimate of the surface reflectance when the AOD is high. Since accurately estimating the surface reflectance is the biggest challenge to overcome when retrieving AOD over land from radiance measurements, I feel that there should be a sentence describing how GOCI handles the estimation of surface reflectance over land. Although probably too much to ask, I would have been more interested in (the diurnal variation of) AOD from the Advanced Himawari Imager, which seems more suited to measuring AOD over land with its shortwave IR channel. I suggest that the authors assess whether 68% of GOCI-AERONET AOD differences fall within the $0.15\tau \pm 0.05$ accuracy envelope mentioned on P8L11 by lumping together the AERONET-GOCI pairs from all sites during KORUS-AQ (especially if the authors feel this does not repeat previous work).

The authors found that WRF-Chem AOD does not correlate well with AERONET AOD, which is interesting, albeit disappointing. WRF-Chem's AOD bias is smaller than that of GOCI AOD however.

Specific comments:

P1L24 (see also P17L10): "the deficiency to describe" -> "a deficiency in describing"

P1L24: relatively -> relative

P1L27: patter -> pattern

P1L27: relative -> relatively

P1L29: "for constrain" -> "to constrain"

P2L1: "Their presence leads to" -> "They are involved in"

P2L5: track -> tract

P2L9: averages -> average

P2L10: with -> and

P2L17: "24" -> "24 times"

P2L17: This sentence implies that clouds do not exist at the surface. However fog is essentially cloud at the surface. Please verify that fog does not affect the performance of surface aerosol monitors.

P2L21: Delete "in emerging need of surface monitors"

P3L5: "one of its first kind" -> "the first of its kind"

P3L5: "that, through" -> "involving"

P3L11: Delete "needed"

P3L12: that -> what

P3L32: Add "over water" after hourly data.

P4L5: " "season invariant" " -> "seasonally invariant"

P4L14: Delete "studying".

P4L14: Add "studies" after variation.

P4L24: Re "since these concentrations", please clarify whether the concentrations were higher at midnight in fall/winter relative to spring/spring or in general (i.e., averaged over day).

P5L2: add space before $m^{-3}$.

P5L3: peaks -> enhancements

P5L8: emergently -> urgently

P5L16: category -> categories          (?)

P5L16: What does "with an accuracy of 90%" mean? If this means "typical errors of ±10%", please use this expression instead.

P6L5: advances-> improvements

P6L28: Delete "Joint"

P6L28: "between the" -> "developed at"

P6L29: Delete "data"

P7L14: Cite reference for CREATE if possible.

P8L9: Imagine -> Imaging

P8L11: Move "compared to V1" immediately after "bias" in this line.

P8L18: Define HUFS here.

P8L28: Delete "(or baseline)". I suggest "climatological" over "baseline" throughout the manuscript.

P8L29: form -> from

P9L18: Do the authors mean: "has fewer intra-class similarities as compared to the coastal rural class."?

P9L29: "evening-build" -> "late afternoon buildup"   (see also P10L8)

P10L14: "maxima AOD" -> "AOD maxima" (see also P10L15).

P10L19 (and elsewhere): Exponent -> exponent

P10L20: This sentence is debatable. It implies that naturally occurring smoke from biomass burning is a minor contributor to the aerosol load. If that is true, the statement is OK.

P10L27: build -> increase

P10L32: build -> rise

P11L11: statically -> statistically

P11L16: later -> late

P11L17: started -> start

P11L17: diminishes -> diminish

P12L13: c -> cohesive              (?)

P12L28: "findings surprising" -> "surprising finding"

P12L28: "longer observation than the" -> "as long a record as"

P13L3 (and elsewhere): AEROENT -> AERONET   (e.g., see P15L2)

P13L4: time -> times

P13L8: is -> are

P13L12: under-predicted -> under-predict

P13L13: of -> to

P13L26: "over predicted" -> "overpredicted"      ("overestimated" is actually better since these are not predictions, they are measurements)

P13L28: amply -> amplify

P13L28: Can't the larger standard deviation for GOCI AOD be due to larger random errors than AERONET AOD?

P13L32: "minutely" -> "every minute"

P14L13-14: I am not convinced that the minima and maxima are real. The authors should add the standard error as a vertical error bar to each hour of the Jeju curve in Fig. 7c and use these (or a t-test) to judge whether these maxima and minima are statistically significant.

P14L21: Re: "process", is there only one process? If not, change to "processes".

P14L29: The systematic bias is more like 0.04-0.09 between 9 and 15 KST. This needs to be corrected.

P14L32: Delete "also".

P14L32: shows -> show

P14L32 (and elsewhere): later -> late

P15L3: "that from" -> "with"

P15L4: relatively -> relative

P15L6: "too much skewness" -> "an overly strong tendency"                (a suggestion)

P15L11: "Similar as its AOD variables" -> "Similar to AOD"

P15L12: "$PM_{2.5}$" $\rightarrow$ "$PM_{2.5}$ data"

P15L24: outliner -> outlier

P16L4: AEROENET is not the correct spelling.

P17L1: larger -> large

P17L6: prolife -> profile

P28: Is what follows the '±' for y and x the standard deviation of the mean? Please add this info to Fig. 5 caption. Assuming it is one standard deviation, it shows that GOCI AOD has a significant overall low bias.

P29: The caption should be "…AERONET AOD vs WRF-Chem AOD…", etc.

P30: (Fig. 7 caption): m-3 -> $m^{-3}$                (occurs twice)

P30: (Fig. 7 caption, and elsewhere): PM2.5 -> $PM_{2.5}$

P30: (Fig. 8 caption): R2 -> $R^{2}$

P31: I suggest plotting AERONET and GOCI separately in Fig. 9b.

---

## Referee Comment (RC2) · Anonymous Referee #3 · 3 Jul 2018

This study discussed the diurnal variations of AOD and PM2.5 in South Korea based on the Aeronet, satellite (GOCI), KORUS-AQ observation and WRF-Chem model. Although the authors highlighted the diurnal variations of AOD and PM2.5, the scientific questions are not mentioned in the whole manuscript especially in introduction. I believe the parameters of diurnal variations of AOD and PM2.5 may be useful for the assessment of aerosol radiative forcing, but this study is out of this topic. Moreover, there is nothing new findings of this manuscript and the ACPD revision was similar with the original version with few revision. I wish the authors would address the followed

critical comments and carefully polish the English throughout the manuscript. 1. I suggest the authors to provide the progress of relevant studies in South Korea rather than USA in the section of introduction or the background and motivation. 2. Since the results from WRF-Chem were poorly matched with the observation, why the authors still used it? What can we learn from it?

---

## Referee Comment (RC3) · Anonymous Referee #1 · 13 Jul 2018

This study focuses on the KORUS-AQ field campaign over South Korea, and also did a lot of statistical analysis of satellite data, meso-scale numerical modeling, AERONET and other ground based sensors. It adds valuable information for air quality study over East Asia, especially the climatology of AOD and PM2.5 diurnal variation. Overall it is easy for readers to follow. I agree this paper is accepted. But there also needs some revising of writing.

Specific comments:

[Figure]

1. Please checks through all the abbreviations, and introduces the full name at the first appearance. For this manner, the Abstract should be treated independent from the main body.

2. Is there any diurnal variation studies for angstrom exponent in the literature?

3. Hourly timing are presented in different formats, so please be consistent with the format. It is better to use 14:00 or 18:00 regardless KST or UTC.

4. Page 7-8, three sections of '3.2', please modifies them.

5. Page 14, Line 3, please specify the 'short timeframe' means within 2 months or how long?

6. Figure 9a, please use solid line to represent one variable (AOD) and dashed line to represent the other variable (PM2.5).

---

## Author Response (AR1)

Reply to reviews #1

This study focuses on the KORUS-AQ field campaign over South Korea, and also did a lot of statistical analysis of satellite data, meso-scale numerical modeling, AERONET and other ground based sensors. It adds valuable information for air quality study over East Asia, especially the climatology of AOD and PM2.5 diurnal variation. Overall it is easy for readers to follow. I agree this paper is accepted. But there also needs some revising of writing.

Reply. Thank you for your review.

Specific comments:

1. Please checks through all the abbreviations, and introduces the full name at the first appearance. For this manner, the Abstract should be treated independent from the main body.

Reply. Thanks. In the abstract, we've expanded acronyms for AERONET, GOCI, KORUS-AQ, and WRF-Chem, and PM2.5. In the main text, we also expanded GMS, GOES, GEOS-Chem, WRF-Chem, PM2.5, and AERONET.

2. Is there any diurnal variation studies for angstrom exponent in the literature?

Reply. Yes. but only few. The following is added into the text. "While AOD diurnal variation has been analysed by several past studies, few studies examined the diurnal variation of Angstrom exponent. Wang et al. (2014) showed that diurnal variation of Angstrom exponent in average has a diurnal variation of 30% (with minima at mid-afternoon) in the dust source region of Gobi desert. Globally, Kaufman et al. (2000) showed the ratio of Angstrom exponent at Terra satellite overpass time with respect to the daily mean is close 1 in 60% of days for the AERONET sites in 1993-1999, and they clearly showed that the diurnal variation of Angstrom exponent is much larger than the counterpart of AOD. Recently, Song et al. (2018) also showed that the diurnal variation of Angstrom exponent is ~15% in southwest China with minima at mid-afternoon, and less than 10% in northern China plains.".

3. Hourly timing are presented in different formats, so please be consistent with the format. It is better to use 14:00 or 18:00 regardless KST or UTC.

Reply. Good point. In several places, we have changed 9 pm to 21:00 KST, for example.

4. Page 7-8, three sections of '3.2', please modifies them.

Reply. Good catch. Done.

5. Page 14, Line 3, please specify the 'short timeframe' means within 2 months or how long?

Reply. Text was added to clarify that we are referring to the six-week KORUS-AQ Field Campaign.

6. Figure 9a, please use solid line to represent one variable (AOD) and dashed line to represent the other variable (PM2.5).

Reply. Done.

Review of Lennartson et al. (submitted to ACP)

Overall comments:

I recommend this paper for publication after some suggested revisions. This is an interesting study that explores different dimensions (spatial variations, diurnal variations, etc.).

**Reply.** Thank you for your time in doing a constructive, thorough review that is now considered in our revision to improve the manuscript.

The paper clearly needs to be improved in terms of use of English. Some corrections that I suggested at quick-review stage were not included in the current manuscript. The corrections or additions that were already made to the manuscript will be helpful for the readers.

**Reply.** Thanks. We have done more proof-reading for the revised manuscript.

GOCI does not have a channel longer than 865 nm. This seems like a weakness since it must be difficult to retrieve an unbiased estimate of the surface reflectance when the AOD is high. Since accurately estimating the surface reflectance is the biggest challenge to overcome when retrieving AOD over land from radiance measurements, I feel that there should be a sentence describing how GOCI handles the estimation of surface reflectance over land. Although probably too much to ask, I would have been more interested in (the diurnal variation of) AOD from the Advanced Himawari Imager, which seems more suited to measuring AOD over land with its shortwave IR channel. I suggest that the authors assess whether 68% of GOCI-AERONET AOD differences fall within the 0.15   ± 0.05 accuracy envelope mentioned on P8L11 by lumping together the AERONET-GOCI pairs from all sites during KORUS-AQ (especially if the authors feel this does not repeat previous work).

**Reply.** While GOCI doesn't have NIR band, it also used a different approach to tread surface reflectance. In the revision we have added that: "The V2 algorithm (Choi et al., 2018) uses the climatology of land surface reflectance that is obtained from the minimum reflectivity technique; in this technique, the minimum value of multi-year top-of-the-atmosphere reflectance measured by GOCI (for each pixel, each month and each hour) after Rayleigh correction is considered as the surface reflectance (for that pixel, that month, and that hour)".  The 0.15tau ± 0.05 accuracy is based on the 5-year validation of GOCI retrieval (Choi et al., 2018). We added that "When validating with AERONET AOD from 2011 to 2016, the V2 reduced median bias compared to V1, and 62% and 71% of GOCI-AERONET AOD difference is within the expected error (EE) of AOD retrieved from MODIS dark target(DT) algorithm over land and ocean, respectively.".  The AERONET-GOCI plot during KORUS-AQ is shown in Figure 5.  We added that "during the KORUS-AQ time period, only ~40% of GOCI-AERONET AOD difference is within EE of MODIS DT over land because of its systematic low bias by 0.04. By adding COGI AOD by 0.04, 40% is increased to 56% (Figure 5c)."

The authors found that WRF-Chem AOD does not correlate well with AERONET AOD, which is interesting, albeit disappointing. WRF-Chem's AOD bias is smaller than that of GOCI AOD however.

**Reply.** Yes, indeed. We revised the text with more description. "WRF-Chem can both over- and under-predict the AERONET AOD values, thereby yielding no bias overall, while the GOCI satellite typically underpredicted them during the KORUS-AQ timeframe (with an overall low bias of 0.04). In reference to

the AERONET AOD values, GOCI AOD has smaller RMSE (0.16) than that (0.28) of WRF-Chem predictions".

Specific comments:

P1L24 (see also P17L10): "the deficiency to describe" -> "a deficiency in describing"

Done.

P1L24: relatively -> relative

Done.

P1L27: patter -> pattern

Done.

P1L27: relative -> relatively

Done.

P1L29: "for constrain" -> "to constrain"

Done.

P2L1: "Their presence leads to" -> "They are involved in"

Done.

P2L5: track -> tract

Done.

P2L9: averages -> average

Done.

P2L10: with -> and

Done.

P2L17: "24" -> "24 times"

Done.

**P2L17: This sentence implies that clouds do not exist at the surface. However fog is essentially cloud at the surface. Please verify that fog does not affect the performance of surface aerosol monitors.**

Reply. We revise the text as follows. "These monitors provide invaluable information regarding $PM_{2.5}$ levels 24 times per day and are not affected by weather conditions particle-bound water included in the sampled air is removed by heating at a constant temperature (usually at 50°C) inside the monitoring instrument (Wang et al., 2006).".

P2L21: Delete "in emerging need of surface monitors"

Done.

P3L5: "one of its first kind" -> "the first of its kind"

Done.

P3L5: "that, through" -> "involving"

Done.

P3L11: Delete "needed"

Done.

P3L12: that -> what

Done.

P3L32: Add "over water" after hourly data.

Done.

P4L5: " "season invariant" " -> "seasonally invariant"

Done.

P4L14: Delete "studying".

Done.

P4L14: Add "studies" after variation.

Done.

P4L24: Re "since these concentrations", please clarify whether the concentrations were higher at midnight in fall/winter relative to spring/spring or in general (i.e., averaged over day).

**Reply.** Done. Text was added to clarify that the concentrations were higher from late afternoon to midnight in fall/winter relative to spring/summer.

P5L2: add space before m-3

Done.

P5L3: peaks -> enhancements

Done.

P5L8: emergently -> urgently

Done.

P5L16: category -> categories (?)

Done.

P5L16: What does "with an accuracy of 90%" mean? If this means "typical errors of ±10%", please use this expression instead.

Done.

P6L5: advances-> improvements

Done.

P6L28: Delete "Joint"

Done.

P6L28: "between the" -> "developed at"

Done.

P6L29: Delete "data"

Done.

P7L14: Cite reference for CREATE if possible.

Added the citation as follows:

**Reply.** Goldberg, D. L., Saide, P. E., Lamsal, L. N., de Foy, B., Lu, Z., Woo, J.-H., Kim, Y., Kim, J., Gao, M., Carmichael, G., and Streets, D. G.: A top-down assessment using OMI NO2 suggests an underestimate in the NOx emissions inventory in Seoul, South Korea during KORUS-AQ, Atmos. Chem. Phys. Discuss., https://doi.org/10.5194/acp-2018-678, in review, 2018.

P8L9: Imagine -> Imaging

Done.

P8L11: Move "compared to V1" immediately after "bias" in this line.

Done.

P8L18: Define HUFS here.

Done.

P8L28: Delete "(or baseline)". I suggest "climatological" over "baseline" throughout the manuscript.

Done.

P8L29: form -> from

Done.

P9L18: Do the authors mean: "has fewer intra-class similarities as compared to the coastal rural class."?

Yes. Thank you for the suggested clarification. The change has been made.

P9L29: "evening-build" -> "late afternoon buildup" (see also P10L8)

Done. The change has been made at P9L29 and P10L8.

P10L14: "maxima AOD" -> "AOD maxima" (see also P10L15).

Done. The change has been made at P10L14 and P10L15.

P10L19 (and elsewhere): Exponent -> exponent

Done. Changes have been made to read "Angstrom exponent" versus "Angstrom Exponent" throughout.

P10L20: This sentence is debatable. It implies that naturally occurring smoke from biomass burning is a minor contributor to the aerosol load. If that is true, the statement is OK.

**Reply.** We added one sentence in the end of this paragraph. "this is especially true for the KORUS-AQ because biomass burning sources in east Asia in growing season (e.g., the study period here) is minimal (Polivka et al., 2015).".

P10L27: build -> increase

Done.

P10L32: build -> rise

Done.

P11L11: statically -> statistically

Done.

P11L16: later -> late

Done.

P11L17: started -> start

Done.

P11L17: diminishes -> diminish

Done.

P12L13: c -> cohesive (?)

Yes. The change has been made.

P12L28: "findings surprising" -> "surprising finding"

Done.

P12L28: "longer observation than the" -> "as long a record as"

Done.

P13L3 (and elsewhere): AEROENT -> AERONET (e.g., see P15L2)

Done. "AEROENT" has been changed to "AERONET" throughout.

P13L4: time -> times

Done.

P13L8: is -> are

Done.

P13L12: under-predicted -> under-predict

Done.

P13L13: of -> to

Done.

P13L26: "over predicted" -> "overpredicted" ("overestimated" is actually better since these are not

predictions, they are measurements)

Done. Over predicted was changed to overestimated.

P13L28: amply -> amplify

Done.

P13L28: Can't the larger standard deviation for GOCI AOD be due to larger random errors than AERONET

AOD?

Yes. Thank you for suggesting this possibility. Text was added.

P13L32: "minutely" -> "every minute"

Done.

P14L13-14: I am not convinced that the minima and maxima are real. The authors should add the

standard error as a vertical error bar to each hour of the Jeju curve in Fig. 7c and use these (or a t-test)

to judge whether these maxima and minima are statistically significant.

**Reply.** The maxima and minima here are indeed not distinct. We added that "the difference between

maxima and minima within 4 µg m$^{-3}$ (or ~10-12% from the mean), suggesting insignificant diurnal

variation pattern".

P14L21: Re: "process", is there only one process? If not, change to "processes".

Done. Changed to "processes."

P14L29: The systematic bias is more like 0.04-0.09 between 9 and 15 KST. This needs to be corrected.

From 9-15 KST, GOCI AOD ranges from 0.25 to 0.3. Perhaps this 0.04-0.09 range would be from 9-16
KST?

Changed. Yes, 0.04 overall low bias is consistent with results in Figure 5, but biases increases in the late afternoon.

P14L32: Delete "also".

Done.

P14L32: shows -> show

Done.

P14L32 (and elsewhere): later -> late

Done. Later was changed to "late" throughout.

P15L3: "that from" -> "with"

Done.

P15L4: relatively -> relative

Done.

P15L6: "too much skewness" -> "an overly strong tendency" (a suggestion)

Thank you for the suggestion. This text was added.

P15L11: "Similar as its AOD variables" -> "Similar to AOD"

Done.

P15L12: "PM2.5" -> "PM2.5 data"

Done.

P15L24: outliner -> outlier

Done.

P16L4: AEROENET is not the correct spelling.

Done.

P17L1: larger -> large

Done.

P17L6: prolife -> profile

Done.

P28: Is what follows the '±' for y and x the standard deviation of the mean? Please add this info to Fig. 5 caption. Assuming it is one standard deviation, it shows that GOCI AOD has a significant overall low bias.

Clarifying text was added.

P29: The caption should be "…AERONET AOD vs WRF-Chem AOD…", etc.

Done.

P30: (Fig. 7 caption): m-3 -> $m^{-3}$ (occurs twice)

Done.

P30: (Fig. 7 caption, and elsewhere): PM2.5 -> PM2.5

Done.

P30: (Fig. 8 caption): R2 -> R2

Done.

P31: I suggest plotting AERONET and GOCI separately in Fig. 9b.

Done.

Reply to reviews #2

This study discussed the diurnal variations of AOD and PM2.5 in South Korea based on the Aeronet, satellite (GOCI), KORUS-AQ observation and WRF-Chem model. Although the authors highlighted the diurnal variations of AOD and PM2.5, the scientific questions are not mentioned in the whole manuscript especially in introduction. I believe the parameters of diurnal variations of AOD and PM2.5 may be useful for the assessment of aerosol radiative forcing, but this study is out of this topic.

Reply. Thank you for your review. The importance of diurnal variation of AOD vs. PM2.5 relationship is discussed in the introduction section and section for background and motivation. As articulated in these sections, there is a growing interesting to derive surface PM2.5 from satellite-based AOD and other ancillary data

Moreover, there is nothing new findings of this manuscript and the ACPD revision was similar with the original version with few revision. I wish the authors would address the followed critical comments and carefully polish the English throughout the manuscript.

1. I suggest the authors to provide the progress of relevant studies in South Korea rather than USA in the section of introduction or the background and motivation.

**Reply.** We added the following into the discussion. If there more suggestions, please us know. "In South Korea, Ghim et al. (2015) showed that PM2.5 average concentration in Seoul in 2002-2008 peaks at 9 am and again around mid-night, but such typical diurnal variation can sometimes be affected by long-range transport of dust.  Similar diurnal variation for PM10 was also found by Yoo et al. (2015) over the S. Korean Peninsular, although its peak at daytime is one hour lagged behind that of PM2.5, which is found in Ghim et al. (2015). Furthermore, both Ghim et al. (2015) and Yoo et al. (2015) showed that PM concentrations can significantly vary with space and time."

Ghim, Y. S., Y.-S. Chang, and K. Jung (2015), Temporal and Spatial Variations in Fine and Coarse Particles in Seoul, Korea, *Aerosol and Air Quality Research*, *15*(3), 842-852, doi:10.4209/aaqr.2013.12.0362.

Yoo, J. M., M. J. Jeong, D. Kim, W. R. Stockwell, J. H. Yang, H. W. Shin, M. I. Lee, C. K. Song, and S. D. Lee (2015), Spatiotemporal variations of air pollutants ($O_3$, $NO_3$, $SO_2$, CO, $PM_{10}$, and VOCs) with land-use types, *Atmos. Chem. Phys.*, *15*(18), 10857-10885, doi:10.5194/acp-15-10857-2015.

2. Since the results from WRF-Chem were poorly matched with the observation, why the authors still used it? What can we learn from it?

**Reply.** We didn't use WRF-Chem for any process studies in this paper. Part of the motivation for this paper is to evaluate the performance of WRF-Chem . We found that WRF-Chem AOD in average has close-to-zero mean bias with respect to AOD measured by AERONET (Fig. 5a), although the correlation is only 0.4. This low correlation is in part because the model lacks the fidelity for describing diurnal variation of AOD, which suggests future improvement for WRF-Chem. Thereby, the observation data can be helpful for improving the model prediction. We discussed those in the summary section of this paper.

**Diurnal variation of aerosol optical depth and PM2.5 in South Korea: a synthesis from AERONET, satellite (GOCI), KORUS-AQ observation, and WRF-Chem model**

Elizabeth Lennartson[1], Jun Wang[1], Juping Gu[2], Lorena Castro Garcia[1], Cui Ge[1], Meng Gao[1,3], Myungje Choi[4], Pablo Saide[5], Gregory R. Carmichael[1], Jhoon Kim[4], Scott Janz[6]

[1]Department of Chemical and Biochemical Engineering, Center for Global and Regional Environmental Research, University of Iowa, U.S.A.
[2]Department of Electrical Engineering, Nantong University, China
[3]School of Engineering and Applied Sciences, Harvard University, U.S.A.
[4]Department of Atmospheric Science, Yonsei University, Republic of Korea
[5]Department of Atmospheric & Oceanic Sciences, University of California - Los Angeles, U.S.A.
[6]Lab for Atmospheric Chemistry and Dynamics, Code 614, NASA Goddard Space Flight Center, U.S.A.

*Correspondence to*: Jun Wang (jun-wang-1@uiowa.edu), Elizabeth Lennartson (elizabeth-lennartson@uiowa.edu)

**Abstract.** Spatial distribution of diurnal variations of aerosol properties in South Korea, both long term and short term, is studied by using 9 AERONET (AErosol RObotic NETwork) sites from 1999 to 2017 and an additional 10 sites during the KORUS-AQ (KORea U.S.-Air Quality) field campaign in May and June of 2016. The extent to which WRF-Chem (Weather Research and Forecasting coupled with Chemistry) model and the GOCI (Geostationary Ocean Color Imager) satellite retrieval can describe these variations is also analyzed. In daily average, Aerosol Optical Depth (AOD) at 550 nm is 0.386 and shows a diurnal variation of 20 to -30% in inland sites, respectively larger than the counterparts of 0.308 and ± 20% in coastal sites. For all the inland and coastal sites, AERONET, GOCI, and WRF-Chem, and observed PM2.5 (Particulate Matter with aerodynamic diameter less than 2.5 μm) data generally show dual peaks for both AOD and PM2.5, one in the morning (often at ~ 8:00-10:00 KST, especially for PM2.5) and another in the early afternoon (~14:00 KST, albeit for PM2.5 this peak is a smaller and sometimes insignificant peak). In contrast, Angstrom exponent values in all sites are between 1.2 and 1.4 with the exception of the inland rural sites having smaller values near 1.0 during the early morning hours. All inland sites experience a pronounced increase of Angstrom exponent from morning to evening, reflecting overall decrease of particle size in daytime. To statistically obtain the climatology of diurnal variation of AOD, a minimum of requirement of ~2 years of observation is needed in coastal rural sites, twice more than the urban sites, which suggests that diurnal variation of AOD in urban setting is more distinct and persistent. While Korean GOCI satellite retrievals are able to consistently capture the diurnal variation of AOD (albeit its systematically low bias of 0.04 in average and up to 0.09 in later afternoon hours), WRF-Chem clearly has a deficiency in describing the relative change of peaks and variations between the morning and afternoon, suggesting further studies for the diurnal profile of emissions. Furthermore, the ratio between PM2.5 and AOD in WRF-Chem

is persistently larger than the observed counterparts by 30-50% in different sites, but no consistent diurnal variation pattern of this ratio can be found. Overall, the relative small diurnal variation of $PM_{2.5}$ is in high contrast with large AOD diurnal variation, which suggests the large diurnal variation of AOD-$PM_{2.5}$ relationships (with $PM_{2.5}$/AOD ratio being largest in the early morning, decreasing around noon, and increasing in late afternoon) and therefore, the need to use AOD from geostationary satellites to constrain either modeling or analysis of surface $PM_{2.5}$ for air quality application.

**1 Introduction**

[revised manuscript text omitted]

1. What is the climatology of AOD and its wavelength-dependence (Angstrom exponent) diurnal variation in South Korea, both spatially and spectrally? How long should the ground measurement record be to derive the climatology of AOD diurnal variation?

20 2. To what degree can AOD diurnal variation be captured by GOCI (Geostationary Ocean Color Imager) and a chemistry transport model WRF-Chem (Weather Research and Forecasting coupled with Chemistry)?

3. What is the diurnal variation of surface $PM_{2.5}$? How well is the diurnal variation of $PM_{2.5}$ – AOD relationship captured by WRF-Chem?

The rest of the paper is organized as follows: Section 2 gives a brief overview of previous studies and the motivation for

25 this research. Section 3 details the datasets used in this study, and Section 4 contains the methods and analysis of the study. Section 5 closes the paper with a summary and the main conclusions.

**2 Background and motivation**

**2.1 Diurnal Variation AOD and Angstrom Exponent**

The study of AOD diurnal variation dates back to the late 1960s but did not gain momentum until near the turn of the

30 century (Barteneva et al. 1967; Panchenko et al. 1999; Peterson et al. 1981; Pinker et al. 1994). Peterson et al. (1981) found the AOD at Raleigh, North Carolina to have an early afternoon maxima at 13:00-14:00 KST during the 1969-1975 study period. Pinker et al. (1994) showed that AOD in sub-Saharan Africa increased throughout the day in December 1987 while

the January 1989 data showed a maxima at 13:00 KST and minima at 10:00 and 16:00 KST. As recent as the early 2000s, the science community agreed that the "diurnal effects are largely unknown and little studied due to the paucity of data…" (Smirnov et al. 2002).

Most diurnal variation of AOD research stemmed from the analysis of aerosol radiative forcing which requires the knowledge of the diurnal distribution of key aerosol properties such as AOD, the single scattering albedo, and the asymmetry factor (Kassianov et al. 2013; Kuang et al. 2015; Wang et al. 2003b). Two early studies developed an algorithm to retrieve AOD diurnal variation from geostationary satellites over water and showed strong AOD diurnal variation during long-range aerosol transport events; Wang et al. (2003b) used April 2001 hourly data over water from the GMS (Geostationary Meteorological Satellite) imager and Wang et al. (2003a) used half-hourly data in July 2000 from the GOES 8 (Geostationary Operational Environmental Satellite) satellite during the ACE-Asia and PRIDE (PuertoRIco Dust Experiment) campaigns, respectively. Consistent with AERONET observations, the GOES 8 retrieval over Puerto Rico showed the dust AOD diurnal variation's noontime minimum and early morning or late afternoon maximum. Subsequent work by Wang et al. (2004) investigated the Taklimakan and Gobi dust regions in China using 1999-2000 AOD data from a nearby airport's sun photometer. They found a "seasonally invariant" diurnal change of more than ±10% for dust AOD. Their results aligned with similar past studies which found the diurnal variation of dust aerosols to be ± <5-15% depending on the AERONET site's location and distance from a dust source region (Kaufman et al. 2000; Levin et al. 1980; Wang et al. 2003a). However, on a daily basis, the day-to-day variation of AOD can be distinct, up to 150% and both daily diurnal variation changes and relative departures of AOD from the daily mean are of up to 20% (Kassianov et al. 2013; Kuang et al. 2015). Furthermore, using long-term Sun-photometer data from China Aerosol Remote Sensing Network (CARSNET), Song et al. (2018) showed that AOD diurnal variation can be up to 30% with peak around noon in northwest China and a steady increase of 40% from early morning to late afternoon in the northern China plains.

While AOD diurnal variation has been analysed by several past studies, few studies examined the diurnal variation of Angstrom exponent. Wang et al. (2014) showed that diurnal variation of Angstrom exponent in average has a diurnal variation of 30% (with minima at mid-afternoon) in the dust source region of Gobi desert. Globally, Kaufman et al. (2000) showed the ratio of Angstrom exponent at Terra satellite overpass time with respect to the daily mean is close 1 in 60% of days for the AERONET sites in 1993-1999, and they clearly showed that the diurnal variation of Angstrom exponent is much larger than the counterpart of AOD. Recently, Song et al. (2018) also showed that the diurnal variation of Angstrom exponent is ~15% in southwest China with minima at mid-afternoon, and less than 10% in northern China plains.

Overall, research based on limited ground-based observations has shown that on a global and annual scale, the AOD diurnal variation exists, albeit relatively small. On a daily and local scale, AOD and Angstrom exponent diurnal variations are significant which calls upon the need of geostationary satellite measurements for both air quality and climate studies. Much less studied is the diurnal variation of Angstrom exponent. Newer geostationary satellites may play an important role for the future generation of AOD diurnal variation studies.

**2.2 PM$_{2.5}$ Diurnal Variation**

In addition to AOD diurnal variation, studies have also investigated the diurnal variation of PM$_{2.5}$. Epidemiological studies focused on the mass, size, spatial and temporal variability, and chemical composition of PM to investigate the complex sources and evolution of aerosols in the atmosphere (Fine et al. 2004; Sun et al. 2013; Wittig et al. 2004). In many of these studies, tracer species of primary aerosols and possible components of secondary organic aerosols were the main focus. (Edgerton et al. 2006; Querol et al. 2001; Sun et al. 2013; Wittig et al. 2004).

Regarding diurnal variation of PM$_{2.5}$ mass, studies have found different results for various locations around the world. Querol et al. (2001) used data from June 1999-June 2000 and found Barcelona, Spain's diurnal variation of PM$_{2.5}$ in all four seasons to be characterized by an increase from the late afternoon to midnight. This trend was more pronounced in winter and autumn since the concentrations were higher from late afternoon to midnight in fall/winter relative to the spring/summer values.

In the United States, early studies have focused on the Los Angeles, Pittsburgh, and general southeast US areas. Fine et al. (2004) chose two sites, an urban one located at the University of Southern California (USC) and a rural one in Riverside, and studied the diurnal variation for one week in the summertime and one week in the wintertime. The USC site had a summer peak in the morning and midday with a winter peak in the morning. The Riverside site experienced a summer peak in the morning and a winter peak in the overnight hours. The winter results were attributed to the boundary-layer temperature inversion that forms throughout the day over the area. A few years later in Pittsburgh, Wittig et al. (2004) found no clear PM$_{2.5}$ diurnal variability due to the combined effect of particulate matter species being transported to the area versus generated locally. Additionally, they concluded that the daily changes in PM$_{2.5}$ concentrations could be "attributed to the major components of the [particulate] mass, namely the sulfate." Data from the 1998-1999 Southeastern Aerosol and Characterization Study (SEARCH) was used by Edgerton et al. (2006) at four pairs of urban-rural sites. They established the following three main PM$_{2.5}$ temporal variation patterns: large values of > 40-50 $\mu$g m$^{-3}$ that occurred on time scales of a few hours, buildup occurring over several days and then returning to normal levels, and enhancements during the summer of similar magnitude as the monthly or quarterly averages. Their four sites had similar diurnal variations characterized by maxima at 6:00-8:00 KST, and again from 18:00-21:00 KST, similar to those results found by Wang and Christopher (2003) at seven sites in Alabama. In South Korea, Ghim et al. (2015) showed that PM$_{2.5}$ average concentration in Seoul in 2002-2008 peaks at 9:00 am and again around mid-night, but such typical diurnal variation can sometimes be affected by long-range transport of dust. Similar diurnal variation for PM$_{10}$ was also found by Yoo et al. (2015) over the S. Korean Peninsular, although its peak at daytime is one hour lagged behind that of PM2.5, which is found in Ghim et al. (2015). Furthermore, both Ghim et al. (2015) and Yoo et al. (2015) showed that PM concentrations can significantly vary with space and time. Hence, 
[revised manuscript text omitted]
 (Choi et al., 2018) uses the climatology of land surface reflectance that is obtained from the minimum reflectivity technique; in this technique, the minimum value of multi-year top-of-the-atmosphere reflectances measured by GOCI (for each pixel, each month and each hour) after Rayleigh correction is considered as the surface reflectance (for that pixel, that month, and that hour). The V2 algorithm has shown similar AOD at 550 nm as

Moderate Resolution Imaging Spectroradiometer (MODIS) and Visible Infrared Imaging Radiometer Suite (VIIRS). When validating with AERONET AOD from 2011 to 2016, the V2 reduced median bias compared to V1, and 62% and 71% of GOCI-AERONET AOD difference is within the expected error(EE) of MODIS dark target(DT) over land and ocean, respectively. 
[revised manuscript text omitted]

20   daily mean ranges from ± 8%.

      Figure 9b displays the diurnal variation of the PM$_{2.5}$/AOD ratio derived from WRF-Chem and collocated AERONET AOD vs. in situ PM$_{2.5}$ ratio and GOCI AOD vs. in situ PM$_{2.5}$ ratio measurement throughout the KORUS-AQ campaign. The comparison of PM$_{2.5}$/AOD ratio is valuable because satellite AOD often is used to multiply this ratio to derive surface PM$_{2.5}$. Overall, the WRF-Chem's PM$_{2.5}$/AOD ratio (with a mean of 154 μg m$^{-3}$ τ$^{-1}$) is 30-50% larger than and shows temporal

25   disparity with the observed counterparts (with a mean of 110 and 103 μg m$^{-3}$ τ$^{-1}$, respectively); it shows a steady decrease from morning to the late afternoon, while the observation-based ratios (based on GOCI or AERONET AOD vs. in situ PM$_{2.5}$) are consist with each other in terms of mean (with correlation of 0.63, Fig. 9b) – they first decrease in the morning, reach the minimum in late morning (11 KST), and then increase steadily toward late afternoon. However, there is no apparent trend between PM$_{2.5}$/AOD ratio and time of day at induvial sites (Fig. 10), and the correlation of the ratio between

30   the WRF-Chem and those observed by AERONET and in situ (Fig. 10) can vary from 0.28 to 0.78, depending on the specific location of each site (Fig. 10). Except Daejeon for some hours and one outlier at a particular hour in Gwangju site (Fig. 10), all other sites show that WRF-Chem's PM2.5/AOD ratio is larger than observation-based counterparts, with the majority of the ratios ranging from 60-140 μg m$^{-3}$ τ$^{-1}$ with outliers as low as 40 and as high as 160 μg m$^{-3}$ τ$^{-1}$ (Fig. 10). The

three coastal rural sites of Baengnyeong, Jeju, and Ulsan have ratio maximums in both the morning (7, 8, and 10 KST) and early evening (17 KST). Their minimums range from morning (9 KST), noontime, and late afternoon (16 KST). The two coastal urban sites of Bulkwang and Busan show more similarities with peaks in the early morning (8 and 9 KST) but still have a minimum range from noontime to afternoon (12 and 15 KST). The inland urban sites have morning (9 and 10 KST),

5    afternoon (13 KST), and early evening (17 and 18 KST) maximums but cohesively have a 15 KST minima, aside from Gwangju whose ratio steadily increases after 10 KST. Being the only inland rural site, the PM$_{2.5}$/AOD at HUFS has a maxima at 8 KST and steadily decreases afterward for the remainder of the day. Consequently, given the large spatial and temporal variations of PM$_{2.5}$/AOD ratio, diurnal variation of AOD from geostationary satellite can be an integral part for deriving PM$_{2.5}$ values from AOD.

10   **4 Summary and conclusions**

By using all possible AERONET data in South Korea, the surface observation of PM$_{2.5}$, GOCI AOD and WRF-Chem simulated AOD during KORUS-AQ Field Campaign in South Korea from April to June 2016, this study analyzed the diurnal variation of aerosol properties and surface PM$_{2.5}$ from surface observations, and assessed their counterparts from models. In summary, the following were found.

15   1.   Long-term AERONET data shows that the climatological AOD diurnal variation is very similar amongst South Korean AERONET sites. Most see an AOD maxima in the middle morning (10 am) and middle afternoon (2 pm) and a noontime AOD minima. Additionally, the coastal sites have lower average values near 0.3 at 550 nm while the inland sites have higher values near 0.4. The inland sites also experience the most AOD fluctuations during the day on the order of +20% to -30%. Analysis of the Angstrom exponent shows a gradual increase throughout the day

20      from 1.2 to 1.4.

   2.   Given there is a persistent diurnal variation of AOD and Angstrom exponent in South Korea, we analyzed that at minimum, there should be more than 12 months of observation, and the coastal rural sites require twice of observations than the inland urban sites, to characterize the climatology of diurnal variation of AOD at statistically significant level. This suggests the distinct and persistent diurnal variation of aerosol properties in urban areas.

25   3.   The AERONET and GOCI AOD had a linear correlation coefficient of (R) 0.8 and RMSE = 0.16 while the AERONET and WRF-Chem relationship had R = 0.4 and RMSE = 0.28, suggesting that AOD data retrieved from GOCI satellite shows a closer agreement with AERONET AOD data than those from WRF-Chem model.

   4.   Analysis of 10 AERONET-surface PM$_{2.5}$ paired sites show that the diurnal variation of PM$_{2.5}$ was ~10% throughout the day, with the exception of the Daejeon and HUFS sites having a maxima at 8 KST (or peaks by 20%) and values

30      gradually decreasing and remaining steady for the remainder of the day after 12 KST. PM$_{2.5}$ daily-mean values were around 30 $\mu$g m$^{-3}$ which is still 20 $\mu$g m$^{-3}$ below the 24-hour PM$_{2.5}$ air quality standard in South Korea but 5 $\mu$g m$^{-3}$ above the WHO recommendation. Overall, the day-to-day variation of mean PM$_{2.5}$ at all sites can be best described

by the variation of hourly PM$_{2.5}$ data at noontime for each day, and is least captured by the variation of PM$_{2.5}$ in the mid-night hours.

5. AERONET, GOCI, WRF-Chem, and observed PM$_{2.5}$ data consistently show dual peaks for both AOD and PM$_{2.5}$, one at 10 KST and another that 14 KST. However, WRF-Chem show the peak in afternoon is larger than the peak in the morning, which is opposite from what GOCI and AERONET reveal. Consequently, WRF-Chem shows increase of AOD and PM$_{2.5}$ from 9 KST to 16 KST, which contrasts with the deceasing counterparts in GOCI, AERONET, and observed PM$_{2.5}$. The analysis suggests that the diurnal profile of emissions in WRF-Chem may have a too large skewness toward the afternoon.

6. PM$_{2.5}$/AOD ratio ranged from 60-120 $\mu$g m$^{-3}$ $\tau^{-1}$ throughout the day, and no consistent pattern was seen at each of 10 sites nor when further broken down into land classification. However, overall, the ratio in WRF-Chem is persistently larger than the observed counterparts by 30-50%, and the observed ratio (from AERONET vs. in situ) shows lower values in late morning and afternoon. In contrast, the diurnal variation of observed ratio is in better agreement with the ratio from GOCI vs. in situ, both showing the minimum at 11 KST and an increase toward late afternoon. This highlights the need to use geostationary satellite to characterize AOD as a function of time as an integrate part toward improved estimate of surface PM$_{2.5}$.

By using rich data sets during KORUS-AQ, this study revealed there are persistent diurnal variation of AOD and surface PM$_{2.5}$ in South Korea. It is shown that the Korean GOCI satellite is able to consistently capture the diurnal variation of AOD, while WRF-Chem clearly has the deficiency to describe the relatively change in the morning and afternoon. As a minimum of one-year observation is required to fully characterize the climatology of diurnal variation pattern of AOD, future field campaigns are commended to have at least longer time periods of surface observations where AERONET and surface PM$_{2.5}$ network can be collocated. Hence, future studies are needed to evaluate the statistical significance of our analysis of diurnal variation of PM$_{2.5}$/AOD ratios with a longer record of observation data.

**Acknowledgement**

This manuscript was part of author Lennartson's master's thesis, and was in part supported by NASA's GEO-CAPE program and in part by KORUS-AQ program (grant #: NNX16AT82G). AERONET data are downloaded from http://aeronet.gsfc.nasa.gov and we thank all AERONET PIs in South Korea for collecting the data. GOCI aerosol product was supported by the "Development of the integrated data processing system for GOCI-II" funded by the Ministry of Ocean and Fisheries, Korea. E. Lennartson and J. Wang also thank Prof. Charles Stanier in the University of Iowa for his constructive comments.

[revised manuscript text omitted]

PM during the KORUS-AQ Campaign
01 May 2016 to 15 June 2016

20   **Figure 8. The diurnal variation of $R^2$ for $PM_{2.5}$ for the sites in Table 2.**

[Figure]

PM during the KO
01 May 2016 t

[Figure]

**Figure 9.** (a) The diurnal variations for AERONET, WRF-Chem, GOCI, and PM$_{2.5}$ for the sites in Table 2 using temporally and spatially matched data. (b) The diurnal variation of WRF-Chem PM$_{2.5}$ /AOD ratio and observation-based counterparts (e.g., GOCI AOD/in situ PM$_{2.5}$ and AERONET AOD/in situ PM$_{2.5}$ ratio) for the average of the sites in Table 2 using temporally and spatially matched data.  Also shown in the legend on the right-hand size of (b) is the mean value of these ratios, and the correlation of GOCI /in situ and WRF-Chem hourly PM2.5/AOD ratios with respect to the that of AERONET /in situ, respectively.

[Figure]

[Figure]

Figure 10. The diurnal variation of the observed (AERONET vs. in situ) and WRF-Chem PM$_{2.5}$ /AOD ratio for each site in Table 2 using temporally and spatially matched data. The ratio is in unit of µg m$^{-3}$ $\tau^{-1}$.